# Exploring Mode Connectivity in Krylov Subspace for Domain Generalization

**Aodi Li**[1], **Liansheng Zhuang**[1,2(✉)], **Xiao Long**[1], **Houqiang Li**[2] **& Shafei Wang**[3]
[1]School of Cyber Science and Technology, University of Science and Technology of China
[2]National Engineering Laboratory for Brain-Inspired Intelligence Technology and Applications, University of Science and Technology of China    [3]Peng Cheng Laboratory, Shenzhen, China
`aodili@mail.ustc.edu.cn, lszhuang@ustc.edu.cn`

## Abstract

This paper explores the geometric characteristics of loss landscapes to enhance domain generalization (DG) in deep neural networks. Existing methods mainly leverage the local flatness around minima for improved generalization. However, recent theoretical studies indicate that flatness does not universally guarantee better generalization. Instead, this paper investigates a global geometrical property for domain generalization, i.e., *mode connectivity*, the phenomenon where distinct local minima are connected by continuous low-loss pathways. Different from flatness, mode connectivity enables transitions from poor to superior generalization models without leaving low-loss regions. To navigate these connected pathways effectively, this paper proposes a novel Billiard Optimization Algorithm (BOA), which discovers superior models by mimicking billiard dynamics. During this process, BOA operates within a low-dimensional Krylov subspace, aiming to alleviate the curse of dimensionality caused by the high-dimensional parameter space of deep models. Furthermore, this paper reveals that oracle test gradients strongly align with the Krylov subspace constructed from training gradients across diverse datasets and architectures. This alignment offers a powerful tool to bridge training and test domains, enabling the efficient discovery of superior models with limited training domains. Experiments on DomainBed demonstrate that BOA consistently outperforms existing sharpness-aware and DG methods across diverse datasets and architectures. Impressively, BOA even surpasses the sharpness-aware minimization by 3.6% on VLCS when using a ViT-B/16 backbone.

## 1 Introduction

Understanding the geometry of loss landscapes (Li et al., 2018a; Xu et al., 2024) in deep neural networks has emerged as a powerful and insightful approach for interpreting the generalization behavior of these models (Wu et al., 2017; Rangamani et al., 2020). Recent empirical and theoretical studies (Keskar et al., 2016; Dziugaite & Roy, 2017; Jiang et al., 2019) have demonstrated a strong connection between generalization performance and the geometric properties of the loss surface, particularly its

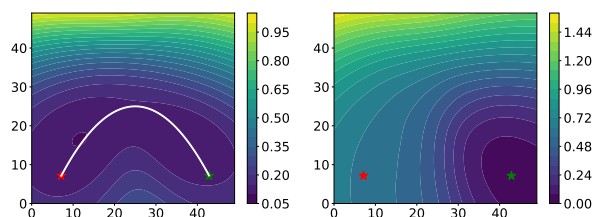

(a) **Training Loss Landscape**    (b) **Test Loss Landscape**

Figure 1: **Ideal and non-ideal models in the training and test loss landscapes.** (a) In the training loss landscape, the ideal (★) and non-ideal (★) models are connected via a low loss path. (b) The ideal model (★) is located in the basin of the test loss landscape, while the non-ideal model (★) is not.

sharpness. It has been observed that flat minima sought by sharpness-aware minimization (SAM) tend to generalize better than sharp ones (Foret et al., 2020; Kwon et al., 2021; Kim et al., 2022). This underscores the potential of leveraging insights from loss geometry.

Although numerous flatness-seeking methods have been developed to learn generalizable models (Kwon et al., 2021; Kim et al., 2022; Zhang et al., 2024), emerging research (Zhang et al., 2021;

Table 1: **Comparisons with other optimization algorithms in update subspaces and rules.** The update direction of SGD/SGLD lies in a random subspace $\mathcal{R}$; SAM lies in a two-dimensional Krylov subspace $\mathcal{K}_2$; and our BOA lies in a $K$-dimensional Krylov subspace $\mathcal{K}_K$.

| Methods | Involving Subspaces | Update Rules |
|---|---|---|
| SGD/SGLD (Welling & Teh, 2011) | $\mathcal{R} = \{g, z\}, \quad z \sim \mathcal{N}(\mathbf{0}, \boldsymbol{I})$. | $\Delta\boldsymbol{\theta} = -\epsilon g + \sqrt{2\epsilon\tau}z$ |
| SAM (Foret et al., 2020) | $\mathcal{K}_2 = \{g, \boldsymbol{H}g\}, \quad \boldsymbol{H}g \approx \frac{\nabla\ell(\boldsymbol{\theta}+\epsilon g) - \nabla\ell(\boldsymbol{\theta})}{\epsilon}$. | $\Delta\boldsymbol{\theta} = -\eta\nabla\ell(\boldsymbol{\theta} + \epsilon g)$ |
| **BOA (Ours)** | $\mathcal{K}_K = \{g, \boldsymbol{H}g, \ldots, \boldsymbol{H}^{K-1}g\}$. | $\begin{cases} \boldsymbol{p}_0 = -\sum_{k=0}^{K-1} \boldsymbol{H}^k g, \\ \boldsymbol{p}_i = \left(\boldsymbol{I} - 2\frac{(\text{proj}_{\mathcal{K}_K}g)(\text{proj}_{\mathcal{K}_K}g)^\top}{\|\text{proj}_{\mathcal{K}_K}g\|^2}\right)\boldsymbol{p}_{i-1} \end{cases}$ |

Wen et al., 2023) indicates that flatness does not universally guarantee improved generalization. This reveals limitations in over-relying on local sharpness measures, particularly within challenging Domain Generalization (DG) settings. In the DG task, our goal is to learn a model from multiple source domains and generalize it to unseen target domains with different data distributions (Wang et al., 2022; Zhou et al., 2021; Shen et al., 2021). In such scenarios, the connection between flatness and generalization becomes significantly more complex and context-dependent. Crucially, the relationship between flatness in parameter space and the geometry of the representation space suggests that low sharpness may not adequately capture a model's vulnerability to domain shifts in feature space (Andriushchenko et al., 2023). Thus, although pursuing flat minima can be beneficial, it should not be considered a unique solution for DG. It is essential to investigate other geometric properties of the loss landscape beyond flatness to achieve better generalization.

In contrast to local flatness, this paper focuses on a global geometric property termed *mode connectivity* (Freeman & Bruna, 2016; Garipov et al., 2018; Draxler et al., 2018). That is, distinct local minima, discovered via independent training from different initializations, are connected by continuous pathways of low loss. This paper empirically investigates this phenomenon in the context of domain generalization (DG) and observes a similar connectivity: As visually illustrated in Figure 1, basins corresponding to models with significantly different out-of-domain performance (e.g., a non-ideal model with poor DG accuracy and an ideal model with nearly 100% DG accuracy) are connected via low-loss pathways. It suggests that the loss landscape is not composed of isolated basins but exhibits a connected structure among separate solutions. This connectivity implies a promising possibility: transitioning from a solution with poor generalization properties to one with strong out-of-domain performance without escaping the low-loss region. Despite its theoretical promise, effectively navigating these pathways in high-dimensional parameter spaces remains challenging due to the curse of dimensionality. Conventional optimization methods, such as SGD and its Langevin dynamics extensions (Bussi & Parrinello, 2007; Welling & Teh, 2011), are often ineffective in this context, as they tend to become trapped in local regions (Deng et al., 2020; Zheng et al., 2024), preventing the discovery of superior solutions. This problem motivates the need for a more deliberate algorithm capable of actively navigating these low-loss tunnels.

Inspired by the dynamics of billiard motion (Bunimovich, 2007; Gutkin, 2003), this paper proposes a novel and efficient traversal algorithm, Billiard Optimization Algorithm (BOA), which explicitly encourages movement along these connected pathways for improved domain generalization. Specifically, BOA operates through two core operations: (1) *line search* to locate loss contour boundaries (analogous to a billiard ball approaching a cushion), and (2) *reflection* to redirect the optimization trajectory upon boundary contact (mimicking momentum-preserving bounces). During this process of high-dimensional optimization, the curse of dimensionality profoundly impacts BOA's performance in two key ways. First, high-dimensional theory (Vershynin, 2018) suggests that random vectors become nearly orthogonal to the optimal initial directions (such as directions of oracle negative test gradients), making naive random direction search highly inefficient in locating useful paths. Second, trajectory sparsity emerges in high-dimensional landscapes, meaning that numerous optimization steps are required to achieve sufficient domain generalization. These issues are compounded in deep models, where the parameter space can be overwhelmingly large. Fortunately, a key geometric regularity identified in our empirical studies offers a solution: the test gradient exhibits strong alignment with the Krylov subspace (Liesen & Strakos, 2013) derived from training gradients across different datasets and architectures. The alignment effectively bridges the gap between training and test domains, and provides near-optimal initial directions for BOA without requiring access to test data. Furthermore, by leveraging this alignment, BOA can constrain its traversal trajectory to a reduced subspace of merely 5-20 dimensions, drastically reducing the vast search space and enabling efficient discovery of models with superior domain generalization capabilities. Empiri-

cal validation conducted on the challenging DomainBed (Gulrajani & Lopez-Paz, 2020) benchmark with various vision transformer architectures demonstrates that BOA consistently outperforms popular sharpness-aware methods and other prevalent DG techniques across five diverse datasets. These results underscore the effectiveness of BOA in navigating high-dimensional loss landscapes and its practical utility for real-world domain generalization tasks.

Our key contributions can be summarized as follows:

- Mode connectivity is identified in the DG context. It describes a phenomenon whereby continuous low-loss trajectories in the parameter space connect models exhibiting substantially divergent out-of-domain performance.

- A novel and efficient Billiards Optimization Algorithm (BOA) is introduced to advance domain generalization. It promotes navigation along low-loss paths connecting distinct local optima, facilitating the identification of models with enhanced generalization capabilities.

- Our study reveals the notable geometric regularity that test gradients demonstrate significant alignment with the Krylov subspace derived from training gradients, thereby establishing an effective bridge between training and unseen test domains.

- Experiments across diverse architectures confirm that BOA consistently surpasses popular sharpness-aware methods and other DG techniques on five datasets of DomainBed.

## 2 RELATED WORK

Although deep learning has achieved success in many application areas, the loss landscapes of deep neural networks remain inadequately understood. This area constitutes an actively evolving field of research, primarily divided into two distinct categories.

**Local structure.** The first category explores the local structure of minima found by SGD and its variants. Researchers have observed that smaller mini-batch sizes often lead to sharp minima, while larger mini-batch sizes tend to yield flat minima (Keskar et al., 2016). In recent years, numerous studies have established a connection between flatness near minimizers and model generalization (Keskar et al., 2016; Dziugaite & Roy, 2017; Jiang et al., 2019): flat minima exhibit stronger generalization capabilities, whereas sharp minima perform poorly on test datasets. This conclusion has also been extended to out-of-distribution generalization scenarios (Zou et al., 2024), attracting widespread attention in the domain generalization research community and inspiring a series of domain generalization methods that seek flat minima, such as SWAD (Cha et al., 2021), SAM (Foret et al., 2020) and its variants (Wang et al., 2023; Li et al., 2025). Unfortunately, however, flatness does not equate to generalization. It is because: (1) There exist sharp yet well-generalizing models. Dinh *et al.* (Dinh et al., 2017) and Kaiyue Wen *et al.* (Wen et al., 2023) have theoretically and empirically identified minimizers that generalize well despite being very sharp. (2) While the generalization benefits of the flattest minimizers have been theoretically established for simple linear models, this classical result does not readily transfer to standard neural networks, including relatively simple architectures such as two-layer ReLU networks (Wen et al., 2023). Unlike previous DG methods, this paper leverages the connectivity of minimizers and attempts to traverse from any given minimizer to one with strong generalization capability by simulating billiard dynamics.

**Global structure.** The other major category of research focuses on the global structure of minima. Over the past few years, numerous studies have revealed connectivity between minima, known as mode connectivity. Freeman *et al.* (Freeman & Bruna, 2016) theoretically proved that half-rectified single-layer networks are asymptotically connected. Their theoretical results are not easily generalizable to multi-layer networks. In contrast, Garipov *et al.* (Garipov et al., 2018) proposed a simple training procedure that finds pathways with nearly constant accuracy even in various modern state-of-the-art architectures, with only one bend between optima. Draxler *et al.* (Draxler et al., 2018) simultaneously and independently discovered curves connecting local optima in DNN loss landscapes. They adopted a different approach to finding these curves, inspired by the "Nudged Elastic Band" method from quantum chemistry. Recently, mode connectivity is used in algorithm design across multiple application areas, such as machine unlearning (Cheng & Amiri, 2025), model merging (Li et al., 2024), and continual learning (Mirzadeh et al., 2020). Although mode connectivity offers the potential to find models with stronger generalization capabilities continuously, this property has not been effectively utilized in the design of out-of-distribution generalization algorithms

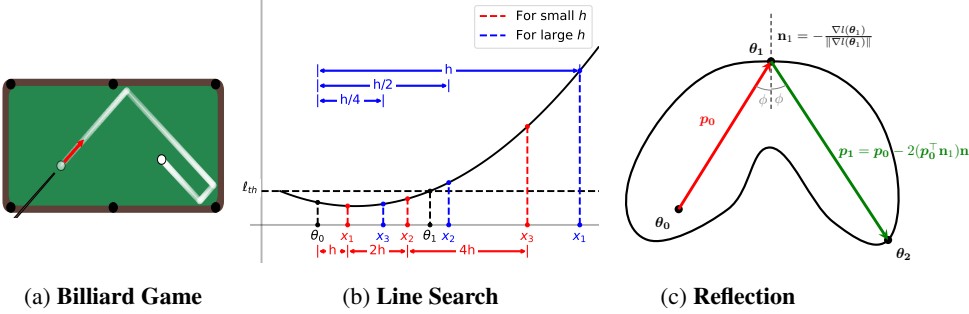

(a) **Billiard Game**           (b) **Line Search**           (c) **Reflection**

Figure 2: **Overview of the proposed Billiard Optimization Algorithm (BOA).** (a) A schematic diagram of a billiards game on a two-dimensional table; (b) The first operation of BOA (Line Search): Like a ball approaching a cushion, this operation identifies loss contour boundaries through line search along directional vectors. (c) The second operation of BOA (Reflection): Upon boundary contact, the trajectory redirects via physics-based rules, mimicking a ball's reflection off a cushion.

due to the curse of dimensionality. To the best of our knowledge, this paper is the first work to seek models with better out-of-domain generalization via mode connectivity.

## 3 METHODOLOGY

Our *Billiard Optimization Algorithm* (BOA) aims to effectively navigate connected pathways within loss landscapes and discover models with superior out-of-domain generalization. Its design draws inspiration from the physical dynamics of billiards, as illustrated in Fig. 2a. Specifically, it iteratively performs two core operations: (1) *line search* to locate the loss contour boundary, analogous to a billiard ball moving toward a cushion (Fig. 2b), and (2) *reflection* to redirect the optimization trajectory upon reaching the boundary, mimicking a ball's reflection off a cushion (Fig. 2c). Due to the space limit, a formalized sketch describing the complete procedure of the Billiard Optimization Algorithm (BOA) is provided in Appendix A.

### 3.1 DEFINITION

In the Billiard Optimization Algorithm (BOA), the conceptual "billiard table" is mathematically defined as the sub-level set of the training loss landscape bounded by a specified contour threshold. Formally, this table corresponds to the parameter domain:

$$\mathcal{T} := \left\{ \boldsymbol{\theta} \in \mathbb{R}^d \mid \ell_{\text{train}}(\boldsymbol{\theta}) \leq \ell_{\text{th}} \right\}, \tag{1}$$

where the contour threshold $\ell_{\text{th}}$ is deliberately constructed as: $\ell_{\text{th}} := \ell_{\text{train}}(\boldsymbol{\theta}_0) + \Delta_\ell$. Here, $\boldsymbol{\theta}_0$ represents the parameter vector of a model trained with an ERM/SAM warm-up, and $\Delta_\ell > 0$ denotes a strictly positive loss increment. This construction guarantees that the warm-up model resides strictly within the interior of the billiard table, satisfying the containment condition: $\ell(\boldsymbol{\theta}_0) < \ell_{\text{th}}$. This foundational definition establishes the optimization landscape as a bounded playfield where subsequent billiard-inspired operations (contour-seeking line searches and momentum-preserving reflections) are performed. The bounded nature of the playfield effectively constrains variations in training loss, ensuring that performance on the training set remains nearly constant while the algorithm searches for model parameters that deliver superior DG performance.

### 3.2 LINE SEARCH

During the line search phase of the Billiard Optimization Algorithm (BOA), we emulate the linear trajectory of a billiard ball approaching a cushion by systematically locating the target loss contour boundary $\ell_{\text{th}} = \ell(\boldsymbol{\theta}_0) + \Delta_\ell$. Unless specified otherwise, we use $\ell$ to represent $\ell_{\text{train}}$ below.

This procedure mathematically corresponds to solving the nonlinear equation with respect to $\alpha$:

$$\ell(\boldsymbol{\theta}_{i-1} + \alpha \boldsymbol{p}_{i-1}) = \ell_{\text{th}}, \tag{2}$$

Table 2: **Out-of-domain accuracies of ViT-B/16 on the DomainBed benchmark.** The star symbol ($\star$) marks experiments involving full fine-tuning of the image encoder, with its absence signifying the application of visual prompt tuning instead.

| Algorithms | VLCS | PACS | OfficeHome | TerraIncognita | DomainNet | Avg. |
|---|---|---|---|---|---|---|
| ERM$^\star$ (Vapnik, 1998) | 81.6 | 93.2 | 80.0 | 54.5 | 57.7 | 73.4 |
| MIRO$^\star$ (Cha et al., 2022) | 83.6 | 95.8 | 82.3 | 58.8 | 57.2 | 75.5 |
| ERM (Jia et al., 2022) | 81.9 | 95.9 | 84.1 | 56.1 | 59.5 | 75.5 |
| IRM (Arjovsky et al., 2019) | 82.9 | 96.1 | 83.2 | 56.7 | 59.1 | 75.6 |
| DANN (Ganin et al., 2016) | 81.8 | 96.3 | 83.0 | 56.0 | 58.4 | 75.1 |
| CDANN (Li et al., 2018c) | 82.4 | 96.5 | 82.9 | 55.6 | 58.4 | 75.2 |
| MMD (Li et al., 2018b) | 82.3 | 95.8 | 83.6 | 57.4 | 59.9 | 75.8 |
| RSC (Huang et al., 2020) | 82.2 | 96.5 | 83.2 | 58.2 | 59.0 | 75.8 |
| CORAL (Sun & Saenko, 2016) | 82.6 | 96.4 | 83.8 | 57.5 | 59.8 | 76.0 |
| IIB (Li et al., 2022b) | 82.3 | 96.5 | 84.2 | 58.2 | 58.6 | 76.0 |
| SAM (Foret et al., 2020) | 82.9 | 96.6 | 85.4 | 56.2 | 59.8 | 76.2 |
| GSAM (Zhuang et al., 2022) | 82.9 | 96.6 | 85.6 | 55.4 | 59.8 | 76.1 |
| GAM (Zhang et al., 2023) | 83.6 | 96.4 | 85.5 | 55.3 | 59.5 | 76.1 |
| SAGM (Wang et al., 2023) | 82.8 | 96.8 | 85.2 | 58.0 | 59.1 | 76.4 |
| DISAM (Zhang et al., 2024) | 82.7 | 97.1 | 85.4 | 57.3 | 59.8 | 76.5 |
| **BOA (Ours)** | **86.5** | **97.4** | **86.0** | **60.3** | **60.2** | **78.1** |

where $\boldsymbol{\theta}_{i-1}$ represents the current model parameter (analogous to the billiard ball's current position) and $\boldsymbol{p}_{i-1}$ denotes the search direction. Candidate points are generated along the ray $\boldsymbol{x}_k = \boldsymbol{\theta}_{i-1} + h_k \boldsymbol{p}_{i-1}$ ($h_k > 0$), with BOA employing an adaptive bracketing strategy to efficiently isolate the solution interval $[h_L, h_R]$: when $\ell(\boldsymbol{x}_k) < \ell_{\text{th}}$, step sizes expand exponentially via $h_{k+1} = (2^k - 1)h$ (where $h$ represents the initial step) until exceeding $\ell_{\text{th}}$; conversely, when $\ell(\boldsymbol{x}_k) > \ell_{\text{th}}$, step sizes contract geometrically through $h_k = h_{k-1}/2$ until falling below $\ell_{\text{th}}$. Following interval isolation, BOA employs the golden-section search method for precise refinement:

$$h_k^{(1)} = h_R - \frac{h_R - h_L}{\psi}, \ h_k^{(2)} = h_L + \frac{h_R - h_L}{\psi}, \tag{3}$$

where $\psi = (1 + \sqrt{5})/2$ represents the golden ratio. The interval boundaries $[h_L, h_R]$ are then updated based on function evaluations at points $h_k^{(1,2)}$. Finally, the approximate optimal solution $\alpha^\star$ of equation (2), found via the golden-section search (Noori et al., 2025), is used to update the parameters:

$$\boldsymbol{\theta}_i = \boldsymbol{\theta}_{i-1} + \alpha^\star \boldsymbol{p}_{i-1}, \tag{4}$$

precisely positioning the model parameter on the loss contour.

### 3.3 REFLECTION

Upon reaching the contour boundary, BOA initiates its second operation: *reflection* that mimics momentum-preserving billiard collisions.

At the contour point $\boldsymbol{\theta}_i$, the reflection direction is computed by leveraging the geometry of the local loss landscape, analogous to the physical reflection process illustrated in Fig. 2c. In this phase, the unit normal vector is expressed as: $\boldsymbol{n}_i = -\frac{\nabla\ell(\boldsymbol{\theta}_i)}{\|\nabla\ell(\boldsymbol{\theta}_i)\|_2}$, which corresponds to the steepest descent direction, plays a role equivalent to the surface normal of the cushion in billiard dynamics. As depicted in Fig. 2c, both the incident direction $\boldsymbol{p}_{i-1}$ and the normal vector $\boldsymbol{n}_i$ define the reflection plane. The specular reflection direction is subsequently updated as follows:

$$\boldsymbol{p}_i = \boldsymbol{p}_{i-1} - 2(\boldsymbol{p}_{i-1}^\top \boldsymbol{n}_i)\boldsymbol{n}_i = (\boldsymbol{I} - 2\boldsymbol{n}_i\boldsymbol{n}_i^\top)\boldsymbol{p}_{i-1}. \tag{5}$$

Geometrically, this transformation subtracts twice the orthogonal projection of $\boldsymbol{p}_{i-1}$ onto $\boldsymbol{n}_i$. This operation preserves the magnitude of momentum, i.e., $\|\boldsymbol{p}_i\|_2 = \|\boldsymbol{p}_{i-1}\|_2$, and adheres to the reflection law $\phi_{\text{incident}} = \phi_{\text{reflected}}$, which implies $(\boldsymbol{p}_{i-1}, \boldsymbol{n}_i) = (\boldsymbol{p}_i, \boldsymbol{n}_i)$. The reflection operator $\boldsymbol{R}[\boldsymbol{n}_i] = \boldsymbol{I} - 2\boldsymbol{n}_i\boldsymbol{n}_i^\top$ represents an improper rotation (with $\det(\boldsymbol{R}) = -1$) that incorporates curvature information via $\boldsymbol{n}_i \propto \nabla\ell(\boldsymbol{\theta}_i)$. This allows efficient exploration along the loss contour without

the need for Hessian recomputation, thereby maintaining a computational complexity of $\mathcal{O}(d)$ per iteration for $d$-dimensional parameters.

After multiple iterations of alternating line search and reflection operations, we generate a trajectory of model parameters and select an optimal model from this trajectory via a validation set for testing.

### 3.4 Overcoming the "Curse of Dimensionality"

The curse of dimensionality fundamentally impacts optimization in high-dimensional parameter spaces through two primary mechanisms: (1) In billiard dynamics, the initial incident direction $\boldsymbol{p}_0$ determines the trajectory's ability to find model parameters with better DG performance. Unlike intuitive 2D billiard, the high-dimensional setting induces a geometric constraint: random vectors become nearly orthogonal to the oracle optimal directions (Vershynin, 2018), suggesting that naive random search becomes extremely inefficient in high dimensions. (2) Trajectory sparsity emerges in high-dimensional landscapes, necessitating numerous optimization steps to achieve sufficient out-of-domain generalization.

To overcome the challenges of high-dimensional optimization, we leverage the geometric structure of the Krylov subspace, which is generated from training derivatives:

$$\mathcal{K}_K(\boldsymbol{H}_{\text{train}}, \boldsymbol{g}_{\text{train}}) = \text{span}\left\{\boldsymbol{g}_0,\ \boldsymbol{H}\boldsymbol{g}_0,\ \ldots,\ \boldsymbol{H}^{K-1}\boldsymbol{g}_0\right\}, \tag{6}$$

where $\boldsymbol{g}_0 = \nabla\ell_{\text{train}}(\boldsymbol{\theta}_0)$ and $\boldsymbol{H} = \nabla^2\ell_{\text{train}}(\boldsymbol{\theta}_0)$. Krylov subspace (Liesen & Strakos, 2013) is famous as an efficient, low-dimensional framework to find approximate solutions to high-dimensional linear algebra problems. Here, we use the Krylov subspace concerning the Hessian matrix to capture the dominant curvature information. It provides two critical advantages: First, it enables principled selection of the initial search direction through efficient approximation of test gradients. Second, it constrains the trajectory to a subspace with limited dimensions that already includes model parameters with better generalization.

**Determine the initial incident direction.** Intuitively, in the DG scenario, the negative test gradient $-\nabla\ell_{\text{test}}(\boldsymbol{\theta}_0)$ might serve as a good choice for the initial incident direction. Fortunately, empirical analysis reveals a remarkable geometric regularity: the test gradient $\nabla\ell_{\text{test}}(\boldsymbol{\theta}_0)$ exhibits strong alignment with the Krylov subspace $\mathcal{K}_K(\boldsymbol{H}_{\text{train}}, \boldsymbol{g}_{\text{train}})$. This alignment enables efficient approximation via setting proper values for $\beta_k$:

$$\boldsymbol{p}_0 = -\sum_{k=0}^{K-1} \beta_k \boldsymbol{H}^k \boldsymbol{g}_0 \approx -\nabla\ell_{\text{test}}(\boldsymbol{\theta}_0). \tag{7}$$

Interestingly, experimental evidence demonstrates that setting $\beta_k = 1$ yields a particularly effective direction. This configuration achieves a small angle between $\boldsymbol{p}_0$ and $-\nabla\ell_{\text{test}}(\boldsymbol{\theta}_0)$ across benchmark datasets, effectively overcoming dimensional barriers. To avoid the computational burden of explicit Hessian calculation, we efficiently approximate Hessian-vector products using a finite-difference method:

$$\boldsymbol{H}\boldsymbol{g}_0 \approx \frac{\nabla\ell(\boldsymbol{\theta}_0 + \epsilon\boldsymbol{g}_0) - \nabla\ell(\boldsymbol{\theta}_0)}{\epsilon}, \tag{8}$$

where the scalar $\epsilon$ is a small step size that balances approximation accuracy and numerical error. This method preserves $\mathcal{O}(d)$ complexity while avoiding direct second-order derivative calculations.

**Constrains the trajectory to limited dimensions.** The Krylov alignment mentioned above also means the existence of better out-of-domain (OOD) solutions within the Krylov subspaces. To limit the reflection operation within the Krylov subspace, the reflection direction $\boldsymbol{p}_i$ can be updated as follows:

$$\boldsymbol{p}_i = (\boldsymbol{I} - 2\tilde{\boldsymbol{n}}_i\tilde{\boldsymbol{n}}_i^\top)\boldsymbol{p}_{i-1}, \tag{9}$$

where $\tilde{\boldsymbol{n}}_i = \text{proj}_{\mathcal{K}_K}\boldsymbol{n}_i$ denotes the projection of $\boldsymbol{n}_i$ within the Krylov subspace $\mathcal{K}_K$.

The Krylov alignment constitutes a "free lunch" in high-dimensional optimization, providing near-optimal initial directions without the availability of test data and reducing the search space substantially. This geometric regularity bridges the gap between training and test distributions, enabling effective exploration of loss landscapes for improved domain generalization.

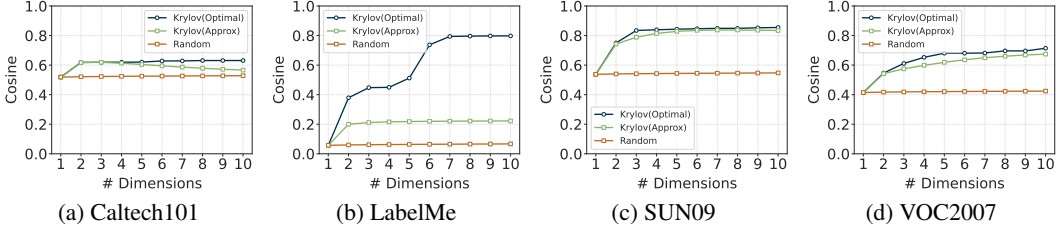

| (a) Caltech101 | (b) LabelMe | (c) SUN09 | (d) VOC2007 |

Figure 3: **Cosine similarity between test gradients and their approximations derived from either random or Krylov subspaces.** Krylov(Optimal): This curve corresponds to the projection of the test gradients onto the Krylov subspace, representing the best possible approximation achievable under the given basis; Krylov(Approx): This curve corresponds to an approximation obtained via the heuristic choice of $\beta_k = 1$ in equation (7); Random: This curve depicts the theoretical expectation value of the projection length of the unit test gradients onto random subspaces (see Appendix B).

## 4 EXPERIMENTS

### 4.1 EXPERIMENTAL SETTING

**Datasets and Protocol.** Our evaluation is conducted on five widely-used, challenging domain generalization datasets: PACS (Li et al., 2017) (9,991 images, 4 domains and 7 classes), VLCS (Fang et al., 2013) (10,729 images, 4 domains and 5 classes), OfficeHome (Venkateswara et al., 2017) (15,588 images, 4 domains and 65 classes), TerraIncognita (Beery et al., 2018) (24,788 images, 4 domains and 10 classes), and DomainNet (Peng et al., 2019) (586,575 images, 6 domains and 345 classes). Following the standardized DomainBed benchmark (Gulrajani & Lopez-Paz, 2020), we implement a rigorous leave-one-domain-out evaluation protocol: each experiment designates one domain as the test set (i.e., test or target domain) while aggregating all remaining domains for training (i.e., training or source domain). To ensure robust model selection and hyperparameter tuning, previous research typically reserves 20% of the data from each source domain to create a validation set. However, we observed that while models may attain similar validation accuracy, their DG performance can differ by over 10% at various points along the training trajectory. It suggests that a low-loss region of the validation set may contain models with widely varying generalization. Thus, relying on this set amounts to random guessing, necessitating the use of a test-domain validation set as a practical alternative. For a fair comparison with prior DG methods, we re-evaluate all baselines using the same test-domain validation set.

**Implementation Details.** Given the demonstrated parameter efficiency and effectiveness of Visual Prompt Tuning (VPT) in domain generalization benchmarks like DomainBed (Jia et al., 2022; Li et al., 2022a; Zheng et al., 2022), our study primarily focuses on VPT-based optimization. To validate the broad applicability of our approach, we conduct comprehensive experiments across three backbones: ViT-B/32, ViT-B/16, and ViT-L/14, all pretrained with CLIP (Radford et al., 2021). Our BOA optimization is applied after a SAM warm-up. During warm-up, we employ a batch size of 16 for the ViT-B models and 8 for ViT-L/14 due to GPU memory constraints, whereas the learning rate is fixed at 5e-4 for all architectures. All experiments employ the Adam optimizer for consistent optimization. During the billiard optimization, we fix the step size $h$ at 10, and set the number of reflections to 20. Other hyperparameters, including the Krylov subspace dimension $K$ and loss variation $\Delta_\ell$, are determined through grid search, with model selection using the validation set.

### 4.2 KRYLOV SUBSPACE ANALYSIS

Prior to the formal evaluation of our BOA algorithm, we present a comprehensive analysis of the Krylov subspace introduced to overcome the curse of dimensionality in high-dimensional optimization.

Firstly, to validate the feasibility of the billiard dynamics, we conduct experiments with oracle test domain: when using the ideal negative test gradient $-\nabla \ell_{\text{test}}(\boldsymbol{\theta}_0)$ as the incident direction, BOA achieves significant performance gains across five datasets (Table 4). The ERM+BOA and SAM+BOA configurations demonstrate consistent improvements averaging +4.9% and +4.8%, re-

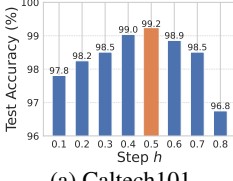 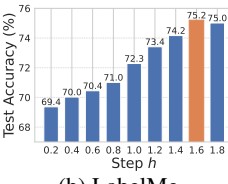 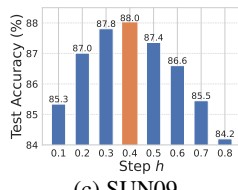 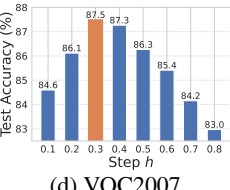

| (a) Caltech101 | (b) LabelMe | (c) SUN09 | (d) VOC2007 |

Figure 4: OOD accuracies along approximate initial incident directions on four VLCS domains.

Table 3: Out-of-domain test accuracies with different ViT backbones.

| Algorithms | VLCS | PACS | OH | TI | DN | Avg. |
|---|---|---|---|---|---|---|
| **Backbone: ViT-B/32** | | | | | | |
| ERM | 82.4 | 94.5 | 79.6 | 36.6 | 54.3 | 69.5 |
| SAM | 81.9 | 95.1 | 80.6 | 42.1 | 54.6 | 70.9 |
| **BOA (Ours)** | **84.9** | **96.3** | **80.9** | **49.7** | **54.9** | **73.3** |
| **Backbone: ViT-B/16** | | | | | | |
| ERM | 81.9 | 95.9 | 84.1 | 56.1 | 59.5 | 75.5 |
| SAM | 82.9 | 96.6 | 85.4 | 56.2 | 59.8 | 76.2 |
| **BOA (Ours)** | **86.5** | **97.4** | **86.0** | **60.3** | **60.2** | **78.1** |
| **Backbone: ViT-L/14** | | | | | | |
| ERM | 82.7 | 98.6 | 90.2 | 61.3 | 64.6 | 79.5 |
| SAM | 82.5 | 98.2 | 90.9 | 64.1 | 64.8 | 80.1 |
| **BOA (Ours)** | **86.4** | **98.7** | **91.2** | **65.7** | **65.4** | **81.5** |

Table 4: Test accuracies when using the oracle negative test gradients as the initial incident directions.

| Algorithms | VLCS | PACS | OH | TI | DN | Avg. |
|---|---|---|---|---|---|---|
| ERM | 81.9 | 95.9 | 84.1 | 56.1 | 59.5 | 75.5 |
| ERM+BOA (Ours) | **85.6** | **97.7** | **86.3** | **69.5** | **62.9** | **80.4** |
| SAM | 82.9 | 96.6 | 85.4 | 56.2 | 59.8 | 76.2 |
| SAM+BOA (Ours) | **87.0** | **97.9** | **87.6** | **69.6** | **62.9** | **81.0** |

Table 5: Comparisons of out-of-domain accuracy when using a ResNet50 architecture.

| Method | Caltech101 | LabelMe | SUN09 | VOC2007 | Avg. |
|---|---|---|---|---|---|
| ERM | 98.2 | 67.1 | 73.0 | 77.6 | 79.0 |
| SAM | 99.6 | 65.7 | 75.1 | 80.5 | 80.2 |
| **BOA** | **99.7** | **69.0** | **77.2** | **81.1** | **81.8** |

spectively, over baseline methods, confirming the feasibility of the billiard dynamics and the ideality of the negative test gradient as the initial incident direction.

Fortunately, we discovered the test gradient exhibits strong alignment with the training-derived Krylov subspace, enabling effective approximation in the absence of test data. Specifically, two facts can be observed in Figure 5: (1) Rapid improvement in directional alignment (quantified by $\cos\gamma_K$) as subspace dimension $K$ increases can be observed, where $\cos\gamma_K$ is defined by the cosine similarity between test gradient and its optimal approximation with the Krylov subspace $\mathcal{K}_K$:

$$\cos\gamma_K = \max_{\boldsymbol{p}\in\mathcal{K}_K} \cos\langle\nabla\ell_{\text{test}}(\boldsymbol{\theta}_0), \boldsymbol{p}\rangle = \cos\langle\nabla\ell_{\text{test}}(\boldsymbol{\theta}_0), \text{proj}_{\mathcal{K}_K}\nabla\ell_{\text{test}}(\boldsymbol{\theta}_0)\rangle, \qquad (10)$$

where $\langle\cdot,\cdot\rangle$ denotes the angle between two vectors. Remarkably, $\cos\gamma_K$ even exceeds 0.8 for $K = 7 \sim 10$ on "LabelMe", which represents an eightfold improvement over the $K = 1$ case ($\cos\gamma_1 < 0.1$). This sharply contrasts with random subspaces, where alignment increases marginally at best. More detailed analysis about Krylov and random subspace can be found in Appendix B. (2) The approximate initial incident direction defined in equation (7) has a similar increasing trend with $K$, achieving near-optimal alignment on "SUN09" and "VOC2007". Note that the approximate initial incident direction is obtained without using test data, different from the optimal approximation that utilizes $\text{proj}_{\mathcal{K}_K}\nabla\ell_{\text{test}}(\boldsymbol{\theta}_0)$. Additionally, we even find a significant increase in test accuracies along the approximate initial incident direction (+1.4% on "Caltech101", +6.5% on "LabelMe", +2.7% on "SUN09" and +2.9% on "VOC2007", see Figure 4). However, this performance gain occurs concurrently with training loss degradation, necessitating BOA's reflection mechanism to maintain the training loss as much as possible while navigating toward generalizable solutions.

## 4.3 MAIN RESULTS

**Comparisons with SOTA domain generalization methods.** Table 2 provides a decisive performance comparison of domain generalization methods on the DomainBed benchmark, clearly establishing our BOA method as the new state-of-the-art. With an average accuracy of 78.1% across all five datasets, BOA outperforms the previous best methods (DISAM at 76.5%) by a significant 1.6% margin. This performance gain represents the largest inter-method improvement observed in the benchmark, demonstrating BOA's exceptional cross-domain generalization capabilities. The dataset-specific analysis reveals particularly striking results; On VLCS, BOA achieves a remarkable 86.5% accuracy, surpassing SAM's 82.9% by a dramatic 3.6% margin. Notably, BOA achieves these record

results using only visual prompt tuning rather than full encoder fine-tuning, making its efficiency particularly impressive. BOA's consistent leadership from easier domains like PACS to highly difficult ones like TerraIncognita demonstrates its effective handling of diverse domain shifts.

**Experiments with different Vision Transformers.** As shown in Table 3, the proposed BOA method consistently outperforms both ERM and SAM across all evaluated Vision Transformer backbones (ViT-B/32, ViT-B/16, ViT-L/14), achieving the highest average accuracy on multi-domain benchmarks. Under the ViT-B/32 backbone, BOA attains an average of 73.3%, outperforming SAM (70.9%) and ERM (69.5%) by margins of 2.4% and 3.8%, respectively, with particularly notable gains on challenging domains such as TI (49.7% vs SAM's 42.1%). This advantage is maintained with larger backbones: using ViT-B/16, BOA reaches 78.1% on average, surpassing SAM by 1.9% and ERM by 3.6%, and with ViT-L/14, it achieves 81.5%, exceeding SAM by 1.4% and ERM by 2.0%. The performance improvement is consistent not only in the aggregate metric but across all five individual datasets, underscoring BOA's robustness and its capability to enhance generalization irrespective of the model capacity or architectural variant of the backbone network.

**Broad applicability to CNN architectures.** While much of our analysis has focused on Vision Transformer architectures, we now demonstrate that the core findings can extend effectively to standard CNN backbones. To this end, we conduct additional experiments using a ResNet50 backbone to validate the persistence of Krylov alignment in CNN architectures. As summarized in Table 6, the superiority of Krylov subspaces over random subspaces remains consistent across multiple datasets, similar to the trends observed in ViT architectures (as illustrated in Figure 3). Notably, on the SUN09 dataset, Krylov alignment (quantified by $\cos \gamma_K$) exceeds 0.75 for $K = 10$, marking an improvement of approximately +0.5 over the $K = 1$ case. This suggests that Krylov subspaces capture geometrically meaningful directions in the loss landscape, even in CNNs.

Table 6: **Cosine similarity between test gradients and their approximations derived from either random or Krylov subspaces (using a ResNet50 backbone).**

| Domain | Subspace Type | Subspace Dimension ($K$) | | | | |
|---|---|---|---|---|---|---|
| | | 2 | 4 | 6 | 8 | 10 |
| Caltech101 | Random | 0.3121 | 0.3122 | 0.3124 | 0.3125 | 0.3125 |
| | **Krylov** | **0.3317** | **0.3339** | **0.3360** | **0.3381** | **0.3381** |
| LabelMe | Random | 0.4063 | 0.4064 | 0.4065 | 0.4066 | 0.4067 |
| | **Krylov** | **0.4937** | **0.5235** | **0.5724** | **0.5825** | **0.5855** |
| SUN09 | Random | 0.2653 | 0.2654 | 0.2655 | 0.2657 | 0.2657 |
| | **Krylov** | **0.4199** | **0.7547** | **0.7603** | **0.7610** | **0.7617** |
| VOC2007 | Random | 0.1395 | 0.1396 | 0.1397 | 0.1399 | 0.1399 |
| | **Krylov** | **0.1406** | **0.2991** | **0.3260** | **0.3414** | **0.3707** |

We also evaluate the DG performance of our BOA method using a ResNet50 backbone. As illustrated in Table 5, BOA achieves the highest accuracy across all four domains, with an average accuracy of 81.8%, outperforming both ERM (79.0%) and SAM (80.2%). The consistency in performance gains reinforces BOA's utility as an architecture-agnostic optimization framework. Notably, in our ResNet experiments, we observed that most hyperparameters could be directly transferred from ViT-B/16 setups. However, due to differences in loss landscape geometry (e.g., CNNs may exhibit sharper minima), we found that maintaining the step size $h = 10$ requires excessive line search iterations to identify solution intervals. To ensure efficient optimization, we adjusted the step size to $h = 1$ for ResNet50.

## 4.4 ABLATION STUDY

Figure 5 presents a decisive ablation study on one VLCS domain ("LabelMe"), evaluating the contribution of Krylov subspace components to the Billiard Optimization Algorithm (BOA) across three loss variation thresholds ($\Delta_\ell = 0.05, 0.1, 0.3$). In Figure 5, Krylov-inspired $\boldsymbol{p}_0$ outperforms random initialization by 0.5-1.6% across all $\Delta_\ell$ values, confirming the effectiveness of Krylov subspace on the determination of $\boldsymbol{p}_0$. Across all three $\Delta_\ell$ values, the complete BOA within Krylov subspace method consistently outperforms intermediate implementations. At $\Delta_\ell = 0.3$, it achieves a remarkable 75.7% accuracy, significantly outperforming both random initialization (69.2%) by 6.5% and

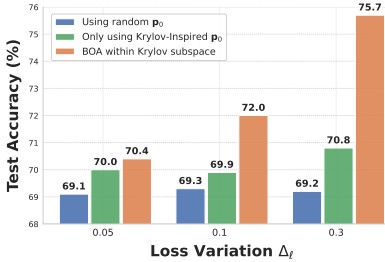

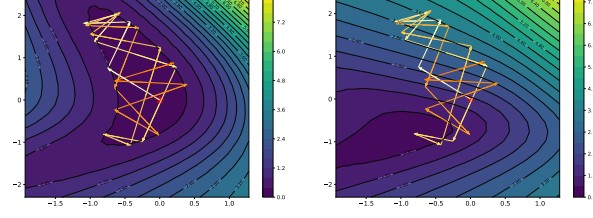

(a) **Training Loss Landscape**   (b) **Test Loss Landscape**

Figure 5: Ablation study on the effect of Krylov subspace using the VLCS domain "LabelMe" for testing.

Figure 6: **Trajectory Visualization of BOA.** The same trajectory, derived from the training loss landscape, is shown in both the (a) training and (b) test loss landscapes.

the standalone Krylov-inspired direction (70.8%) by 4.9%. This substantial margin demonstrates that Krylov-enhanced reflection provides further benefits beyond initial direction selection.

### 4.5 LOSS LANDSCAPE VISUALIZATION

Our loss landscape visualization examines two key aspects: (1) connectivity between models with different out-of-domain performance, and (2) optimization trajectory analysis of the Billiard Optimization Algorithm (BOA).

**Mode Connectivity Analysis**. A baseline model was first trained on three OfficeHome domains ("Clipart", "Product", and "Real World"), which may exhibit poor performance when generalized to the unseen "Art" domain. In contrast, an ideal oracle model was trained using all domains. Subsequently, building upon the approach in (Garipov et al., 2018), we connect these two models using a parameterized Bézier curve and optimize it to maintain uniformly low loss throughout the entire pathway. Finally, as illustrated in Figure 1, a 2D visualization demonstrates a continuous low-loss valley bridging the non-ideal (red marker) and ideal (green marker) models. Remarkably, this connectivity pattern has been consistently observed across diverse datasets and architectures.

**BOA Trajectory Visualization.** To visually demonstrate the effectiveness of the BOA algorithm, we employed constrained Krylov subspaces with a dimension of $K = 2$ to trace its optimization path on the VLCS dataset. As illustrated in Figure 6, BOA successfully navigates the training loss landscape and seeks out-of-domain solutions located within optimal regions on the test loss surface. This visualization offers clear visual evidence of BOA's capability in discovering better models.

## 5 CONCLUSION

This study demonstrates that leveraging mode connectivity in loss landscapes offers a novel and effective approach for enhancing domain generalization. We empirically established that low-loss pathways connect models with divergent generalization capabilities, revealing a geometric structure that enables transitions to superior solutions without leaving low-loss regions. To navigate this structure, we proposed the Billiard Optimization Algorithm (BOA), which efficiently traverses these connected modes using physics-inspired operations. A key insight enabling this efficiency was the strong alignment between test gradients and the low-dimensional Krylov subspace derived from training gradients, allowing BOA to operate effectively in a reduced subspace. Integrated with parameter-efficient vision prompt tuning, BOA achieved superior performance across multiple DomainBed benchmarks, consistently outperforming existing sharpness-aware and domain generalization techniques. These results highlight the value of geometric connectivity properties for developing robust models that generalize well to unseen domains. Future work will explore other geometric properties of loss landscapes or adapt BOA for dynamic environments where domain shifts occur continuously.

### ACKNOWLEDGEMENTS

This work was financially supported by the National Natural Science Foundation of China (NSFC) under Grant No.U20B2070. The authors gratefully acknowledge the NSFC for its critical support of this research.

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
