# SUPPLEMENTARY MATERIAL FOR "EXPLORING MODE CONNECTIVITY IN KRYLOV SUBSPACE FOR DOMAIN GENERALIZATION"

This supplementary file is structured as follows: It begins by presenting the Billiard Optimization Algorithm (BOA) framework in Section A, detailing its two core algorithms for line search and reflection phases within Krylov subspaces to enhance domain generalization. The theoretical foundation is then established through a comprehensive subspace analysis (see Section B), which compares the approximation capabilities of Krylov and random subspaces for test gradients. This includes rigorous mathematical proofs (Theorems B.1, B.8, and Corollary B.9) and supporting lemmas on Gamma/Beta functions in Section B.1. The subsequent Section B.2 provides an empirical investigation into Krylov subspace performance across diverse datasets and architectures, with results visualized in figures. The document concludes with reproducibility details (in Section C), a note on LLM Utilization (in Section D) for language refinement, and a discussion of broader impacts (in Section E).

## A    SKETCH OF BILLIARD OPTIMIZATION ALGORITHM

This section presents a detailed sketch of the Billiard Optimization Algorithm (BOA), a novel optimization technique inspired by the dynamics of billiard motion. It consists of two main algorithms: Algorithm 1 outlines the core BOA framework, which starts with initialization using pre-trained parameters and a loss threshold, then iterates through a line search phase (leveraging Algorithm 2 to find solution intervals and golden-section search for parameter updates) and a reflection phase (computing normal vectors, projecting them into a Krylov subspace, and deriving reflected directions to navigate loss landscapes). Algorithm 2 supports this by efficiently bracketing intervals containing solutions to loss threshold equations. Overall, BOA aims to traverse high-dimensional parameter spaces via physics-inspired operations, enhancing domain generalization by maintaining low loss while exploring connected modes.

---

**Algorithm 1: Billiard Optimization Algorithm**

---

**Input:** Training dataset $\mathcal{D}_{train}$, reflection count $N$, hyperparameters ($K$, $\epsilon$, $\Delta_\ell$, step size $h$).
**Output:** Parameter trajectory $\{\boldsymbol{\theta_i}\}_{i=1}^N$.
1: **Initialization**:
2: Load pretrained parameters $\boldsymbol{\theta}_0$.
3: Compute loss threshold $\ell_{th} \leftarrow \ell(\boldsymbol{\theta}_0) + \Delta_\ell$.
4: Compute initial gradient $\boldsymbol{g}_0 \leftarrow \nabla\ell(\boldsymbol{\theta}_0)$ on $\mathcal{D}_{train}$.
5: Initialize incident direction via the following equation:
$\quad$ $\boldsymbol{p}_0 \leftarrow -\text{normalize}(\sum_{k=0}^{K-1} \boldsymbol{H}^k \boldsymbol{g}_0)$.
6: **for** $i = 1$ **to** $N$ **do**
7: $\quad$ **Line Search Phase**:
8: $\quad$ Determine solution interval $[h_L, h_R]$ via Algorithm 2.
9: $\quad$ Get the approximate solution $\alpha^\star$ of $\ell(\boldsymbol{\theta}_{i-1} + \alpha\boldsymbol{p}_0) = \ell_{th}$ via golden-section search using the above interval.
10: $\quad$ Update the parameters: $\boldsymbol{\theta}_i \leftarrow \boldsymbol{\theta}_{i-1} + \alpha^\star\boldsymbol{p}_{i-1}$.
11: $\quad$ **Reflection Phase**:
12: $\quad$ Compute normal vector: $\boldsymbol{n}_i \leftarrow -\nabla\ell(\boldsymbol{\theta}_i)/\|\nabla\ell(\boldsymbol{\theta}_i)\|$.
13: $\quad$ Project normal vector into the Krylov subspace: $\tilde{\boldsymbol{n}}_i \leftarrow \text{proj}_{\mathcal{K}}\boldsymbol{n}_i$.
14: $\quad$ Compute reflected direction: $\boldsymbol{p}_i \leftarrow (\boldsymbol{I} - 2\tilde{\boldsymbol{n}}_i\tilde{\boldsymbol{n}}_i^\top)\boldsymbol{p}_{i-1}$.
15: **end for**
16: **return** Parameter trajectory $\{\boldsymbol{\theta_i}\}_{i=1}^N$.

---

---

Algorithm 2: Interval Search for Solution Bracket.

---

**Input:** Training dataset $\mathcal{D}_{train}$, current model parameter $\boldsymbol{\theta}$, the incident direction $\boldsymbol{p}$, loss threshold $\ell_{th}$, step size $h$.
**Output:** Interval $[h_L, h_R]$ containing solution to $\ell(\boldsymbol{\theta} + \alpha \boldsymbol{p}) = \ell_{th}$.
1: Initialize $\boldsymbol{x} = \boldsymbol{\theta} + h\boldsymbol{p}$.
2: **if** $\ell(\boldsymbol{x}) < \ell_{th}$ **then**
3:     Initialize $h_L \leftarrow 0$.
4:     **while** $\ell(\boldsymbol{x}) < \ell_{th}$ **do**
5:         $h_L \leftarrow h_L + h; h \leftarrow 2h; \boldsymbol{x} \leftarrow \boldsymbol{x} + h\boldsymbol{p}$.
6:     **end while**
7:     $h_R \leftarrow h_L + h$.
8: **else**
9:     **while** $\ell(\boldsymbol{x}) > \ell_{th}$ **do**
10:         $h \leftarrow h/2; \boldsymbol{x} \leftarrow \boldsymbol{\theta} + h\boldsymbol{p}$.
11:     **end while**
12:     $h_L \leftarrow h; h_R \leftarrow 2h$.
13: **end if**
14: **return** The interval $[h_L, h_R]$.

---

## B    SUBSPACE ANALYSIS

As demonstrated in Table 1 presented in the main manuscript, prevalent optimization algorithms, such as SGD (SGLD Welling & Teh (2011)) and SAM Foret et al. (2020), can uniformly interpret gradient updates through either Krylov subspaces or random subspaces. In this section, we demonstrate the superiority of Krylov subspaces over random subspaces for approximating test gradients. Specifically, we examine the best possible approximation of test gradients achievable when using a set of basis vectors, which mathematically corresponds to projecting the test gradients onto the subspace spanned by those bases.

To analyze random subspaces, we consider the case where the subspace is constrained to include the training gradient, under the rationale that the training gradient may itself be a reasonable approximation to the test gradients. Under this assumption, we derive the expected length of the projection of a unit directional vector onto all such random subspaces that contain the training gradient. For the Krylov subspace, we empirically investigate how the projection length of the test gradients onto the Krylov subspace varies as the dimension of the subspace increases. This dynamic is illustrated through a plotted relationship, highlighting the characteristics of Krylov-based approximation. We also compare this projection length from the Krylov subspace with the expected projection length obtained from random subspaces of the same dimension, clearly showcasing the comparative advantage of Krylov subspaces in capturing the directional information of test gradients more effectively.

### B.1    RANDOM SUBSPACE ANALYSIS

In this section, we focus on analyzing random subspaces that explicitly contain the training gradient vector $\boldsymbol{a}$. We begin by introducing Theorem B.1, which establishes that all such subspaces can be mathematically represented as a direct sum structure: specifically, $\text{span}(\boldsymbol{a}) \oplus W$, where $W$ denotes an arbitrary subspace lying within the orthogonal complement of $\boldsymbol{a}$, i.e., $\text{span}(\boldsymbol{a})^\perp$. This decomposition provides a foundational framework for the subsequent understanding of these constrained random subspaces.

Subsequently, we present Theorem B.8, which offers a detailed statistical characterization (such as the expectation, variance, and concentration bounds) about the projection length of a fixed vector onto random subspaces. To ensure a comprehensive and rigorous derivation of Theorem B.8, we first introduce several essential definitions and preliminary results related to Gamma and Beta functions, along with supporting lemmas. These mathematical tools are critical for facilitating the proofs and subsequent analyses.

By synthesizing the structural insights from Theorem B.1 and the probabilistic findings from Theorem B.8, we derive a significant corollary (Corollary B.9). This corollary delivers an expectation bound for the projection of a fixed vector $\boldsymbol{b}$ (interpreted as the test gradient) onto random subspaces that contain $\boldsymbol{a}$ (interpreted as the training gradient). According to this corollary, we can obtain theo-

retical cosine increases between test gradients and their $n$-dimensional Krylov Subspace projections, which are present in Table 1. Moreover, Corollary B.9 demonstrates a sharp concentration phenomenon, indicating that the probability of the projection magnitude deviating significantly from its expected value is exceedingly low.

**Theorem B.1.** *Let $V$ be a finite-dimensional vector space over a field $\mathbb{F}$ endowed with an inner product $(\cdot, \cdot)$, and let $\boldsymbol{a} \in V$ be a fixed nonzero vector. Define the collection $\mathcal{A}$ as the set of all subspaces of $V$ that contain $\boldsymbol{a}$, that is, $\mathcal{A} = \{U \subseteq V \mid U \text{ is a subspace and } \boldsymbol{a} \in U\}$. Define the collection $\mathcal{B}$ as the set of all subspaces expressible as the direct sum: $\mathrm{span}\,(\boldsymbol{a}) \oplus W$, where $W$ is any subspace of the orthogonal complement $\mathrm{span}(\boldsymbol{a})^{\perp}$ (denoted by $\boldsymbol{a}^{\perp}$ hereafter). Then $\mathcal{A} = \mathcal{B}$.*

*Proof.* To prove $\mathcal{A} = \mathcal{B}$, we demonstrate mutual set inclusion.

First, every subspace $U$ in $\mathcal{B}$ is of the form $\mathrm{span}\,(\boldsymbol{a}) \oplus W$ for some $W \subseteq \boldsymbol{a}^{\perp}$. Since $\boldsymbol{a}$ lies in $\mathrm{span}\,(\boldsymbol{a})$, it is contained in $U$, confirming that $U$ belongs to $\mathcal{A}$. Thus, $\mathcal{B}$ is a subset of $\mathcal{A}$.

Conversely, take any subspace $U$ in $\mathcal{A}$, meaning $\boldsymbol{a} \in U$. Define $W = U \cap \boldsymbol{a}^{\perp}$, which is a subspace of $\boldsymbol{a}^{\perp}$ because it is the intersection of two subspaces. We claim that $U$ decomposes orthogonally as $\mathrm{span}\,(\boldsymbol{a}) \oplus W$. To verify this, observe that for any vector $\boldsymbol{u} \in U$, it can be orthogonally decomposed as $\boldsymbol{u} = \frac{(\boldsymbol{u},\boldsymbol{a})}{(\boldsymbol{a},\boldsymbol{a})}\boldsymbol{a} + \left(\boldsymbol{u} - \frac{(\boldsymbol{u},\boldsymbol{a})}{(\boldsymbol{a},\boldsymbol{a})}\boldsymbol{a}\right)$. The first term is a scalar multiple of $\boldsymbol{a}$ and hence belongs to $\mathrm{span}\,(\boldsymbol{a})$. The second term is orthogonal to $\boldsymbol{a}$ by construction, and since both $\boldsymbol{u}$ and $\boldsymbol{a}$ are in $U$, the second term also lies in $U$; therefore, it belongs to $W$. Moreover, the intersection of $\mathrm{span}\,(\boldsymbol{a})$ and $W$ is trivial: if a vector $c\boldsymbol{a}$ (for some scalar $c$) is in $W$, then it must be orthogonal to $\boldsymbol{a}$, implying $c = 0$ since $\boldsymbol{a} \neq \boldsymbol{0}$. This shows $U \subseteq \mathrm{span}\,(\boldsymbol{a}) \oplus W$. The reverse inclusion is immediate because both $\mathrm{span}\,(\boldsymbol{a})$ and $W$ are contained in $U$. Hence, $U$ is exactly the direct sum $\mathrm{span}\,(\boldsymbol{a}) \oplus W$, and so $U$ is an element of $\mathcal{B}$. This proves $\mathcal{A} \subseteq \mathcal{B}$.

By mutual inclusion, we conclude that $\mathcal{A} = \mathcal{B}$. $\square$

**Definition B.2** (Gamma Function). *The Gamma function, denoted as $\Gamma(z)$, generalizes the factorial function to complex and real numbers (excluding non-positive integers). For $\mathrm{Re}(z) > 0$, it is defined by the improper integral:*

$$\Gamma(z) = \int_0^\infty t^{z-1} e^{-t} dt. \tag{1}$$

*This is also called Euler's integral of the second kind. Key properties of the Gamma Function include:*
*(1) Recurrence relation: $\Gamma(z + 1) = z\Gamma(z)$.*
*(2) For positive integers $n$, $\Gamma(n) = (n-1)!$.*

**Lemma B.3** (Stirling's Approximation for the Gamma Function). *Let $\Gamma(z)$ denote the gamma function defined for $\Re(z) > 0$ by the integral $\Gamma(z) = \int_0^\infty t^{z-1} e^{-t} dt$. As $|z| \to \infty$ in the complex plane with $|\arg z| < \pi$ (i.e., excluding the negative real axis), $\Gamma(z)$ satisfies the asymptotic expansion:*

$$\Gamma(z) \sim \sqrt{\frac{2\pi}{z}} \left(\frac{z}{e}\right)^z \left(1 + \frac{1}{12z} + \frac{1}{288z^2} + \cdots\right). \tag{2}$$

*where the series arises from higher-order corrections in Laplace's method applied to the integral representation of $\Gamma(z)$. The leading term $\sqrt{\frac{2\pi}{z}} \left(\frac{z}{e}\right)^z$ constitutes the basic Stirling approximation, while the subsequent series forms a divergent but asymptotic expansion that converges factorially fast for sufficiently large $|z|$.*

**Definition B.4** (Beta Function). *The Beta function, denoted as $B(x, y)$, is a special function defined for positive real numbers $x > 0$ and $y > 0$ by the integral:*

$$B(x, y) = \int_0^1 t^{x-1} (1-t)^{y-1} dt. \tag{3}$$

*This integral converges for $x > 0$ and $y > 0$, and it is also known as Euler's integral of the first kind. It relates to the Gamma function via the following equation:*

$$B(x, y) = \frac{\Gamma(x)\Gamma(y)}{\Gamma(x+y)}. \tag{4}$$

*It exhibits symmetry (i.e., $B(x, y) = B(y, x)$), and is fundamental in probability theory, particularly for normalizing the Beta distribution.*

**Definition B.5** (Beta Distribution)**.** *The probability density function (PDF) of the Beta distribution is defined for $x \in [0, 1]$ as:*

$$f_X(x) = \frac{1}{B(\alpha, \beta)} x^{\alpha-1}(1 - x)^{\beta-1}, \tag{5}$$

*where $B(\alpha, \beta)$ represents the Beta function.*

**Lemma B.6.** *Let $X$ be a random variable following a Beta distribution with parameters $\alpha > 0$ and $\beta > 0$, denoted as $X \sim Beta(\alpha, \beta)$. The expectation and variance of $\sqrt{X}$ are given by:*

$$\mathbb{E}[\sqrt{X}] = \frac{\Gamma(\alpha + \frac{1}{2})\Gamma(\alpha + \beta)}{\Gamma(\alpha)\Gamma(\alpha + \beta + \frac{1}{2})}, \tag{6}$$

$$\mathbb{V}[\sqrt{X}] = \frac{\alpha}{\alpha + \beta} - \left(\frac{\Gamma(\alpha + \frac{1}{2})\Gamma(\alpha + \beta)}{\Gamma(\alpha)\Gamma(\alpha + \beta + \frac{1}{2})}\right)^2. \tag{7}$$

*Proof.* We derive each component step by step in the following.

**1. Compute the expectation.**

The expectation $\mathbb{E}[\sqrt{X}]$ is computed by integrating $\sqrt{x}$ against the probability density function of the Beta distribution:

$$\mathbb{E}[\sqrt{X}] = \int_0^1 \sqrt{x} f_X(x) dx = \int_0^1 x^{1/2} \cdot \frac{1}{B(\alpha, \beta)} x^{\alpha-1}(1 - x)^{\beta-1} dx. \tag{8}$$

We simplify the integrand to obtain:

$$\mathbb{E}[\sqrt{X}] = \frac{1}{B(\alpha, \beta)} \int_0^1 x^{\alpha + \frac{1}{2} - 1}(1 - x)^{\beta-1} dx. \tag{9}$$

The integral $\int_0^1 x^{\alpha + \frac{1}{2} - 1}(1 - x)^{\beta-1} dx$ corresponds to the Beta function $B\left(\alpha + \frac{1}{2}, \beta\right)$. Therefore, we derive:

$$\mathbb{E}[\sqrt{X}] = \frac{B\left(\alpha + \frac{1}{2}, \beta\right)}{B(\alpha, \beta)}. \tag{10}$$

We express both Beta functions in terms of Gamma functions as follows:

$$B\left(\alpha + \frac{1}{2}, \beta\right) = \frac{\Gamma(\alpha + \frac{1}{2})\Gamma(\beta)}{\Gamma(\alpha + \frac{1}{2} + \beta)}, \quad B(\alpha, \beta) = \frac{\Gamma(\alpha)\Gamma(\beta)}{\Gamma(\alpha + \beta)}. \tag{11}$$

Substituting these expressions into the equation yields:

$$\mathbb{E}[\sqrt{X}] = \frac{\frac{\Gamma(\alpha+\frac{1}{2})\Gamma(\beta)}{\Gamma(\alpha+\frac{1}{2}+\beta)}}{\frac{\Gamma(\alpha)\Gamma(\beta)}{\Gamma(\alpha+\beta)}} = \frac{\Gamma(\alpha + \frac{1}{2})\Gamma(\beta)}{\Gamma(\alpha + \frac{1}{2} + \beta)} \cdot \frac{\Gamma(\alpha + \beta)}{\Gamma(\alpha)\Gamma(\beta)}. \tag{12}$$

The $\Gamma(\beta)$ terms cancel, which leads to the final result:

$$\mathbb{E}[\sqrt{X}] = \frac{\Gamma(\alpha + \frac{1}{2})\Gamma(\alpha + \beta)}{\Gamma(\alpha)\Gamma(\alpha + \beta + \frac{1}{2})}. \tag{13}$$

**2. Compute the variance.**

Define $Y = \sqrt{X}$. The variance of $Y$ is:

$$\mathbb{V}[Y] = \mathbb{E}[Y^2] - (\mathbb{E}[Y])^2 = \mathbb{E}[X] - \left(\mathbb{E}[\sqrt{X}]\right)^2. \tag{14}$$

The expectation of $X$ is:

$$\mathbb{E}[X] = \int_0^1 x \cdot f_X(x)dx = \frac{1}{B(\alpha, \beta)} \int_0^1 x^\alpha (1-x)^{\beta-1}dx. \tag{15}$$

The integral is the Beta function $B(\alpha + 1, \beta)$:

$$\mathbb{E}[X] = \frac{B(\alpha+1, \beta)}{B(\alpha, \beta)} = \frac{\Gamma(\alpha+1)\Gamma(\alpha+\beta)}{\Gamma(\alpha)\Gamma(\alpha+\beta+1)}. \tag{16}$$

Using $\Gamma(z+1) = z\Gamma(z)$, this simplifies to:

$$\mathbb{E}[X] = \frac{\alpha}{\alpha + \beta}. \tag{17}$$

Substitute equation (13) and (17) into the variance formula (14), we obtain:

$$\mathbb{V}[\sqrt{X}] = \frac{\alpha}{\alpha + \beta} - \left(\frac{\Gamma(\alpha+\frac{1}{2})\Gamma(\alpha+\beta)}{\Gamma(\alpha)\Gamma(\alpha+\beta+\frac{1}{2})}\right)^2. \tag{18}$$

This is the exact closed-form expression. $\square$

**Lemma B.7** (Concentration of Lipschitz functions on the sphere Vershynin (2018)). *Let $X$ be a random vector uniformly distributed on the Euclidean sphere of radius $\sqrt{n}$, i.e., $X \sim \mathrm{Unif}(\sqrt{n}S^{n-1})$. Then for any Lipschitz function[1] $f : \sqrt{n}S^{n-1} \to \mathbb{R}$ and any $t \geq 0$, we have*

$$\mathbb{P}\{|f(X) - \mathbb{E}f(X)| \geq t\} \leq 2\exp\left(-\frac{ct^2}{\|f\|_{\mathrm{Lip}}^2}\right). \tag{19}$$

**Theorem B.8.** *Let $\boldsymbol{a}$ be a fixed unit vector in $\mathbb{R}^N$, and let $V$ be an $n$-dimensional subspace of $\mathbb{R}^N$ sampled uniformly from the Grassmannian manifold $G(n, N)$ of all $n$-dimensional subspaces. Let $\boldsymbol{P}_V(\boldsymbol{a})$ denote the orthogonal projection of $\boldsymbol{a}$ onto $V$. Then, we have the following three conclusions: (1) The cosine of the angle $\theta$ between the original vector $\boldsymbol{a}$ and its projection $\boldsymbol{P}_V(\boldsymbol{a})$ satisfies the relation $\cos\theta = \|\boldsymbol{P}_V(\boldsymbol{a})\|$, and the expected value of this cosine scales with the subspace dimension $n$ and the ambient dimension $N$ according to:*

$$\mathbb{E}[\cos\theta] = \mathbb{E}[\|\boldsymbol{P}_V(\boldsymbol{a})\|] = \frac{\Gamma\left(\frac{n}{2}+\frac{1}{2}\right)\Gamma\left(\frac{N}{2}\right)}{\Gamma\left(\frac{n}{2}\right)\Gamma\left(\frac{N}{2}+\frac{1}{2}\right)}, \tag{20}$$

*where $\Gamma$ denotes the gamma function. This expectation increases monotonically with $n$, approaching 1 as $n \to N$. For large $N$ and $n$, $\mathbb{E}[\cos\theta] \approx \sqrt{\frac{n}{N}}$.*
*(2) The variance of the cosine is given by:*

$$\mathbb{V}[\cos\theta] = \mathbb{V}[\|\boldsymbol{P}_V(\boldsymbol{a})\|] = \frac{n}{N} - \left(\frac{\Gamma(\frac{n+1}{2})\Gamma(\frac{N}{2})}{\Gamma(\frac{n}{2})\Gamma(\frac{N+1}{2})}\right)^2, \tag{21}$$

*which exhibits a decay rate of $\mathcal{O}(1/N)$ as $N$ increases, for any fixed dimension $n$.*
*(3) $\cos^2\theta$ is $\mathcal{O}(1/N)$-subgaussian. That is, for all $t > 0$, the tail probability is bounded by:*

$$\mathbb{P}\left(\left|\cos^2\theta - \frac{n}{N}\right| \geq t\right) \leq 2\exp\left(-\frac{t^2}{\mathcal{O}(1/N)}\right). \tag{22}$$

*Proof.* The orthogonal projection $\boldsymbol{P}_V(\boldsymbol{a})$ resides within the subspace $V$, and by the properties of orthogonal projection in inner product spaces, the cosine of the angle $\theta$ between $\boldsymbol{a}$ and its projection is given by $\cos\theta = \frac{(\boldsymbol{a}, \boldsymbol{P}_V(\boldsymbol{a}))}{\|\boldsymbol{a}\|\|\boldsymbol{P}_V(\boldsymbol{a})\|}$. Since $\boldsymbol{a}$ is a unit vector and because $(\boldsymbol{a}, \boldsymbol{P}_V(\boldsymbol{a})) = \|\boldsymbol{P}_V(\boldsymbol{a})\|^2$ (a fundamental property of orthogonal projections), this expression simplifies to $\cos\theta = \|\boldsymbol{P}_V(\boldsymbol{a})\|$. The squared norm $\|\boldsymbol{P}_V(\boldsymbol{a})\|^2$ represents the fraction of the vector's energy preserved in the projection.

---

[1] A function $f : \mathcal{X} \to \mathbb{R}$ is Lipschitz continuous if there exists a constant $\|f\|_{\mathrm{Lip}} \geq 0$ (called the Lipschitz constant) such that for all $x, y \in \mathcal{X}$, $|f(x) - f(y)| \leq \|f\|_{\mathrm{Lip}} \cdot \rho(x, y)$, where $\rho(x, y)$ is a distance metric on the domain $\mathcal{X}$ (e.g., the Euclidean distance in $\mathbb{R}^n$).

Due to the uniform random sampling of $V$ from $G(n, N)$, the quantity $\|\boldsymbol{P}_V(\boldsymbol{a})\|^2$ is a random variable that follows a Beta distribution: $\|\boldsymbol{P}_V(\boldsymbol{a})\|^2 \sim \text{Beta}\left(\frac{n}{2}, \frac{N-n}{2}\right)$.

To determine the distribution of $\|\boldsymbol{P}_V(\boldsymbol{a})\|^2$, we exploit the rotational symmetry of the uniform distribution on $G(n, N)$. Specifically, the uniform distribution on $G(n, N)$ is invariant under the action of the orthogonal group $O(N)$. Thus, without loss of generality, we may fix $\boldsymbol{a} = \boldsymbol{e}_1 = (1, 0, \ldots, 0)^\top$, as the distribution of $\|\boldsymbol{P}_V(\boldsymbol{a})\|^2$ depends only on the geometry of the subspace and not on the specific direction of $\boldsymbol{a}$. Note that a random subspace $V \in G(n, N)$ can be generated by selecting a random orthogonal matrix $Q \in O(N)$ uniformly and defining $V$ as the span of the first $n$ columns of $Q$, denoted $\boldsymbol{q}_1, \ldots, \boldsymbol{q}_n$. The squared projection length is then:

$$\|\boldsymbol{P}_V(\boldsymbol{a})\|^2 = \sum_{i=1}^{n} (\boldsymbol{a}^\top \boldsymbol{q}_i)^2 = \sum_{i=1}^{n} \boldsymbol{e}_1^\top \boldsymbol{q}_i \boldsymbol{q}_i^\top \boldsymbol{e}_1 = \sum_{i=1}^{n} q_{i1}^2, \tag{23}$$

where $q_{i1}$ is the first component of $\boldsymbol{q}_i$. That is, $\|\boldsymbol{P}_V(\boldsymbol{a})\|^2 = \sum_{i=1}^{n} w_i^2$, where $\boldsymbol{w} = (w_1, \ldots, w_N)^\top = Q^\top \boldsymbol{e}_1$ is a random vector uniformly distributed on the unit sphere $S^{N-1}$ in $\mathbb{R}^N$. It is well-known that a random vector $\boldsymbol{w}$ uniformly distributed on $S^{N-1}$ can be generated by normalizing a standard Gaussian vector. Let $X_1, \ldots, X_N$ be independent and identically distributed (i.i.d.) standard normal random variables, $X_i \sim \mathcal{N}(0, 1)$. Then:

$$\boldsymbol{w} = \left(\frac{X_1}{R}, \frac{X_2}{R}, \ldots, \frac{X_N}{R}\right)^\top, \quad \text{where} \quad R = \sqrt{\sum_{j=1}^{N} X_j^2}. \tag{24}$$

Thus, the squared projection becomes:

$$\|\boldsymbol{P}_V(\boldsymbol{a})\|^2 = \sum_{i=1}^{n} w_i^2 = \frac{\sum_{i=1}^{n} X_i^2}{\sum_{j=1}^{N} X_j^2}. \tag{25}$$

Define $S_n = \sum_{i=1}^{n} X_i^2$ and $S_{N-n} = \sum_{k=n+1}^{N} X_k^2$. Since the $X_i$ are i.i.d. $\mathcal{N}(0, 1)$, $S_n \sim \chi^2(n)$ and $S_{N-n} \sim \chi^2(N - n)$ are independent chi-square random variables with $n$ and $N - n$ degrees of freedom, respectively. Consequently:

$$\|\boldsymbol{P}_V(\boldsymbol{a})\|^2 = \frac{S_n}{S_n + S_{N-n}}. \tag{26}$$

The ratio $\frac{S_n}{S_n + S_{N-n}}$ is a Beta-distributed random variable. Specifically, if $A \sim \chi^2(a)$ and $B \sim \chi^2(b)$ are independent, then $\frac{A}{A+B} \sim \text{Beta}\left(\frac{a}{2}, \frac{b}{2}\right)$. Here, $a = n$ and $b = N - n$, so:

$$\frac{S_n}{S_n + S_{N-n}} \sim \text{Beta}\left(\frac{n}{2}, \frac{N-n}{2}\right). \tag{27}$$

Therefore, $\cos^2 \theta = \|\boldsymbol{P}_V(\boldsymbol{a})\|^2$ follows a Beta distribution with parameters $\alpha = \frac{n}{2}$ and $\beta = \frac{N-n}{2}$.

*Proof of (1).* The expected value of the cosine, $\mathbb{E}[\|\boldsymbol{P}_V(\boldsymbol{a})\|]$, is the expectation of the square root of this Beta-distributed variable. For a random variable $X \sim \text{Beta}(\alpha, \beta)$, the expectation $\mathbb{E}[\sqrt{X}]$ is given by $\frac{\Gamma(\alpha+\frac{1}{2})\Gamma(\alpha+\beta)}{\Gamma(\alpha)\Gamma(\alpha+\beta+\frac{1}{2})}$, as depicted in Lemma B.6. Substituting $\alpha = \frac{n}{2}$ and $\beta = \frac{N-n}{2}$ yields the stated result:

$$\mathbb{E}[\|\boldsymbol{P}_V(\boldsymbol{a})\|] = \frac{\Gamma\left(\frac{n}{2} + \frac{1}{2}\right) \Gamma\left(\frac{N}{2}\right)}{\Gamma\left(\frac{n}{2}\right) \Gamma\left(\frac{N}{2} + \frac{1}{2}\right)}. \tag{28}$$

The monotonic increase of $\mathbb{E}[\cos \theta]$ with $n$ follows intuitively because a higher-dimensional random subspace has a greater probability of capturing a significant component of any fixed vector. In the limit as $n$ approaches $N$, the subspace $V$ encompasses the entire ambient space, the projection becomes the identity operation ($\|\boldsymbol{P}_V(\boldsymbol{a})\| = 1$), and thus $\mathbb{E}[\cos \theta] \to 1$. To analyze how $\mathbb{E}(\cos \theta)$ varies with $n$ and $N$, we next apply Stirling's approximation. Stirling's approximation states that for large $z$, $\Gamma(z) \sim \sqrt{\frac{2\pi}{z}} \left(\frac{z}{e}\right)^z$. Applying this to each Gamma function in equation (28) yields:

$$\mathbb{E}[\|\boldsymbol{P}_V(\boldsymbol{a})\|] \approx \frac{\sqrt{\frac{2\pi}{\frac{n}{2}+\frac{1}{2}}} \left(\frac{\frac{n}{2}+\frac{1}{2}}{e}\right)^{\frac{n}{2}+\frac{1}{2}} \cdot \sqrt{\frac{2\pi}{\frac{N}{2}}} \left(\frac{\frac{N}{2}}{e}\right)^{\frac{N}{2}}}{\sqrt{\frac{2\pi}{\frac{n}{2}}} \left(\frac{\frac{n}{2}}{e}\right)^{\frac{n}{2}} \cdot \sqrt{\frac{2\pi}{\frac{N+1}{2}}} \left(\frac{\frac{N+1}{2}}{e}\right)^{\frac{N}{2}+\frac{1}{2}}}. \tag{29}$$

Simplifying radicals and exponents, this reduces to:

$$\mathbb{E}[\|\boldsymbol{P}_V(\boldsymbol{a})\|] \approx \frac{\frac{1}{\sqrt{(n+1)N}}(n+1)^{(n+1)/2}N^{N/2}}{\frac{1}{\sqrt{n(N+1)}}n^{n/2}(N+1)^{(N+1)/2}}. \tag{30}$$

The expression is then algebraically rearranged to isolate dominant terms:

$$\mathbb{E}[\|\boldsymbol{P}_V(\boldsymbol{a})\|] \approx \frac{\sqrt{n(N+1)}}{\sqrt{(n+1)N}} \cdot \frac{\sqrt{(n+1)\cdot\left(1+\frac{1}{n}\right)^n}}{\sqrt{(N+1)\cdot\left(1+\frac{1}{N}\right)^N}}. \tag{31}$$

This rearrangement is exact and leverages properties of exponents and radicals to highlight the asymptotic behavior of the ratio. As $n, N \to \infty$, the terms $\left(1+\frac{1}{n}\right)^n \to e$, $\left(1+\frac{1}{N}\right)^N \to e$. Thus, we have:

$$\mathbb{E}[\|\boldsymbol{P}_V(\boldsymbol{a})\|] \approx \sqrt{\frac{n}{N}}. \tag{32}$$

The final approximation $\sqrt{n/N}$ holds for large $n$ and $N$.

*Proof of (2).* To derive the asymptotic behavior of the variance for the norm of the projection $\|\boldsymbol{P}_V(\boldsymbol{a})\|$, we begin with the exact variance expression obtained from the Beta distribution properties:

$$\mathbb{V}[\|\boldsymbol{P}_V(\boldsymbol{a})\|] = \frac{n}{N} - \left(\frac{\Gamma\left(\frac{n+1}{2}\right)\Gamma\left(\frac{N}{2}\right)}{\Gamma\left(\frac{n}{2}\right)\Gamma\left(\frac{N+1}{2}\right)}\right)^2. \tag{33}$$

This expression arises because $\|\boldsymbol{P}_V(\boldsymbol{a})\|^2 \sim \text{Beta}\left(\frac{n}{2}, \frac{N-n}{2}\right)$, and the variance of $\|\boldsymbol{P}_V(\boldsymbol{a})\|$ is derived from the moments of this distribution. The term $\frac{n}{N}$ corresponds to $\mathbb{E}[\|\boldsymbol{P}_V(\boldsymbol{a})\|^2]$, while the subtracted term is the square of $\mathbb{E}[\|\boldsymbol{P}_V(\boldsymbol{a})\|]$.

To analyze the behavior for large $N$, we apply Stirling's approximation to the Gamma functions. Using Stirling's approximation:

$$\Gamma\left(\frac{N}{2}\right) \approx \sqrt{\frac{4\pi}{N}}\left(\frac{N}{2e}\right)^{N/2}, \tag{34}$$

$$\Gamma\left(\frac{N+1}{2}\right) \approx \sqrt{\frac{4\pi}{N+1}}\left(\frac{N+1}{2e}\right)^{(N+1)/2}. \tag{35}$$

Thus, their ratio simplifies to:

$$\frac{\Gamma\left(\frac{N}{2}\right)}{\Gamma\left(\frac{N+1}{2}\right)} \approx \frac{\sqrt{\frac{4\pi}{N}}\left(\frac{N}{2e}\right)^{N/2}}{\sqrt{\frac{4\pi}{N+1}}\left(\frac{N+1}{2e}\right)^{(N+1)/2}} = \sqrt{\frac{N+1}{N}} \cdot \frac{\left(\frac{N}{2e}\right)^{N/2}}{\left(\frac{N+1}{2e}\right)^{(N+1)/2}}. \tag{36}$$

Simplifying the exponential terms:

$$\frac{\left(\frac{N}{2e}\right)^{N/2}}{\left(\frac{N+1}{2e}\right)^{(N+1)/2}} = \left(\frac{N}{N+1}\right)^{N/2} \cdot \frac{1}{\sqrt{\frac{N+1}{2e}}}. \tag{37}$$

As $N \to \infty$, $\left(1+\frac{1}{N}\right)^{-N/2} \to e^{-1/2}$. Combining all factors:

$$\frac{\Gamma\left(\frac{N}{2}\right)}{\Gamma\left(\frac{N+1}{2}\right)} \approx \sqrt{\frac{N+1}{N}} \cdot \frac{e^{-1/2}}{\sqrt{\frac{N+1}{2e}}} = \sqrt{\frac{2}{N}}. \tag{38}$$

Then, the variance formula contains the squared ratio:

$$\left(\frac{\Gamma\left(\frac{n+1}{2}\right)\Gamma\left(\frac{N}{2}\right)}{\Gamma\left(\frac{n}{2}\right)\Gamma\left(\frac{N+1}{2}\right)}\right)^2 \approx \frac{2}{N}\left(\frac{\Gamma\left(\frac{n+1}{2}\right)}{\Gamma\left(\frac{n}{2}\right)}\right)^2. \tag{39}$$

Thus, the variance becomes:

$$\mathbb{V}\left[\|\boldsymbol{P}_V(\boldsymbol{a})\|\right] \approx \frac{n}{N} - \frac{2}{N}\left(\frac{\Gamma\left(\frac{n+1}{2}\right)}{\Gamma\left(\frac{n}{2}\right)}\right)^2. \tag{40}$$

The term $\left(\frac{\Gamma\left(\frac{n+1}{2}\right)}{\Gamma\left(\frac{n}{2}\right)}\right)^2$ depends only on the fixed dimension $n$, not $N$. Regardless of $n$, this term is bounded by a constant $C_n$. Thus,

$$\mathbb{V}\left[\|\boldsymbol{P}_V(\boldsymbol{a})\|\right] \approx \frac{n}{N} - \frac{C_n}{N} = \mathcal{O}\left(\frac{1}{N}\right). \tag{41}$$

*Proof of (3).* We will prove this conclusion by using Lemma B.7. Now we start with a Lipschitz function.

Consider the function $f : \mathbb{R}^N \to \mathbb{R}$ defined by

$$f(\boldsymbol{x}) = \frac{1}{N}\sum_{i=1}^{n} x_i^2, \tag{42}$$

where $\boldsymbol{x} = (x_1, \ldots, x_N)$ and $n \leq N$. The domain of $f$ is the sphere of radius $\sqrt{N}$ in $\mathbb{R}^N$, denoted $\sqrt{N}S^{N-1}$. This sphere consists of all vectors $\boldsymbol{x} \in \mathbb{R}^N$ with Euclidean norm $\|\boldsymbol{x}\| = \sqrt{N}$, i.e.,

$$\sqrt{N}S^{N-1} = \{\boldsymbol{x} \in \mathbb{R}^N \mid \|\boldsymbol{x}\| = \sqrt{N}\}. \tag{43}$$

This sphere is a Riemannian manifold with intrinsic dimension $N - 1$, and its geometry plays a crucial role in the analysis of concentration phenomena. The function $f$ arises naturally in the context of orthogonal projections: if $\boldsymbol{w} = \boldsymbol{x}/\sqrt{N}$ is a unit vector on $S^{N-1}$, then $f(\boldsymbol{x}) = \sum_{i=1}^{n} w_i^2 = \|\boldsymbol{P}_V(\boldsymbol{a})\|^2$, where $\boldsymbol{P}_V(\boldsymbol{a})$ is the orthogonal projection of a fixed unit vector $\boldsymbol{a}$ onto a random $n$-dimensional subspace $V$ of $\mathbb{R}^N$. To establish the Lipschitz property of the function $f$, let $\boldsymbol{a}, \boldsymbol{b} \in \sqrt{N}S^{N-1}$ be arbitrary points on the sphere. The difference $|f(\boldsymbol{a}) - f(\boldsymbol{b})|$ is given by:

$$|f(\boldsymbol{a}) - f(\boldsymbol{b})| = \left|\frac{1}{N}\sum_{i=1}^{n}(a_i^2 - b_i^2)\right|. \tag{44}$$

Using the identity $a_i^2 - b_i^2 = (a_i - b_i)(a_i + b_i)$ and the triangle inequality, we have:

$$\left|\sum_{i=1}^{n}(a_i^2 - b_i^2)\right| \leq \sum_{i=1}^{n}|a_i - b_i| \cdot |a_i + b_i|. \tag{45}$$

By the Cauchy-Schwarz inequality, we obtain:

$$\sum_{i=1}^{n}|a_i - b_i| \cdot |a_i + b_i| \leq \sqrt{\sum_{i=1}^{n}(a_i - b_i)^2} \cdot \sqrt{\sum_{i=1}^{n}(a_i + b_i)^2}. \tag{46}$$

Since $\|\boldsymbol{a}\|^2 = \|\boldsymbol{b}\|^2 = N$, we have $\sum_{i=1}^{N}a_i^2 = N$ and $\sum_{i=1}^{N}b_i^2 = N$. Thus,

$$\sum_{i=1}^{n}(a_i + b_i)^2 \leq \sum_{i=1}^{N}(a_i + b_i)^2 = \|\boldsymbol{a}\|^2 + \|\boldsymbol{b}\|^2 + 2\boldsymbol{a} \cdot \boldsymbol{b} \leq 2N + 2|\boldsymbol{a} \cdot \boldsymbol{b}| \leq 4N, \tag{47}$$

where the last step follows from $|\boldsymbol{a} \cdot \boldsymbol{b}| \leq \|\boldsymbol{a}\|\|\boldsymbol{b}\| = N$. Combining these, we can get:

$$|f(\boldsymbol{a}) - f(\boldsymbol{b})| \leq \frac{1}{N} \cdot \sqrt{\sum_{i=1}^{n}(a_i - b_i)^2} \cdot \sqrt{4N} = \frac{2}{\sqrt{N}} \cdot \sqrt{\sum_{i=1}^{n}(a_i - b_i)^2}. \tag{48}$$

Finally, $\sqrt{\sum_{i=1}^{n}(a_i - b_i)^2} \leq \|\boldsymbol{a} - \boldsymbol{b}\|$, so

$$|f(\boldsymbol{a}) - f(\boldsymbol{b})| \leq \frac{2}{\sqrt{N}}\|\boldsymbol{a} - \boldsymbol{b}\|. \tag{49}$$

This shows that $f$ is Lipschitz continuous on $\sqrt{N}S^{N-1}$ with Lipschitz constant $L = \frac{2}{\sqrt{N}}$.

Then, according to Lemma B.7, any Lipschitz function concentrates sharply around its mean. Specifically, if $f$ is $L$-Lipschitz, then for a random vector $\boldsymbol{x}$ uniformly distributed on $\sqrt{N}S^{N-1}$, we have

$$\mathbb{P}\left(|f(\boldsymbol{x}) - \mathbb{E}[f(\boldsymbol{x})]| \geq t\right) \leq 2\exp\left(-\frac{ct^2}{L^2}\right), \quad \forall t > 0, \tag{50}$$

where $c > 0$ is an absolute constant . Substituting $L = \frac{2}{\sqrt{N}}$ yields

$$\mathbb{P}\left(|f(\boldsymbol{x}) - \mathbb{E}[f(\boldsymbol{x})]| \geq t\right) \leq 2\exp\left(-\frac{ct^2}{(2/\sqrt{N})^2}\right) = 2\exp\left(-\frac{cNt^2}{4}\right). \tag{51}$$

This bound is valid for all $t \geq 0$, demonstrating that $f(\boldsymbol{x})$ is subgaussian with variance proxy $\sigma^2 = \mathcal{O}(1/N)$.

$\square$

**Corollary B.9.** *Let $V$ be an $N$-dimensional inner product space over $\mathbb{R}$, and let $\boldsymbol{a}, \boldsymbol{b} \in V$ be fixed unit vectors. Define the set of admissible subspaces:*

$$\mathcal{A} := \{U \subseteq V \mid \dim(U) = n, \boldsymbol{a} \in U\}, \quad 1 < n < N. \tag{52}$$

*Let $U$ be a random subspace sampled uniformly from $\mathcal{A}$. Denote by $\boldsymbol{P}_U(\boldsymbol{b})$ the orthogonal projection of $\boldsymbol{b}$ onto $U$, and let $\theta$ be the angle between $\boldsymbol{b}$ and $\boldsymbol{P}_U(\boldsymbol{b})$. Then, we have:*
*(1) Subgaussian concentration:*

$$\mathbb{P}\left(\left|\cos^2\theta - \mathbb{E}[\cos^2\theta]\right| \geq t\right) \leq 2\exp\left(-\frac{t^2}{\mathcal{O}(1/N)}\right), \quad \forall t > 0. \tag{53}$$

*(2) Expectation bound:*

$$\mathbb{E}[\cos\theta] = \mathbb{E}\left[\|\boldsymbol{P}_U(\boldsymbol{b})\|\right] \leq |(\boldsymbol{a}, \boldsymbol{b})| + \frac{\Gamma\left(\frac{n}{2}\right)\Gamma\left(\frac{N-1}{2}\right)}{\Gamma\left(\frac{n-1}{2}\right)\Gamma\left(\frac{N}{2}\right)}. \tag{54}$$

*Proof.* Since $\boldsymbol{a} \in U$, we decompose $\boldsymbol{b}$ orthogonally relative to $\boldsymbol{a}$:

$$\boldsymbol{b} = (\boldsymbol{a}, \boldsymbol{b})\boldsymbol{a} + \boldsymbol{b}_\perp, \quad \boldsymbol{b}_\perp \in \boldsymbol{a}^\perp. \tag{55}$$

According to Theorem B.1, the projection onto $U$ splits into components parallel and orthogonal to $\boldsymbol{a}$:

$$\|\boldsymbol{P}_U(\boldsymbol{b})\|^2 = (\boldsymbol{a}, \boldsymbol{b})^2 + \|\boldsymbol{P}_{U \cap \boldsymbol{a}^\perp}(\boldsymbol{b}_\perp)\|^2, \tag{56}$$

where $U \cap \boldsymbol{a}^\perp$ is a uniformly random $(n-1)$-dimensional subspace in the $(N-1)$-dimensional space $\boldsymbol{a}^\perp$. The term $\|\boldsymbol{P}_{U \cap \boldsymbol{a}^\perp}(\boldsymbol{b}_\perp)\|^2$ is the squared norm of the projection of $\boldsymbol{b}_\perp$ onto a random $(n-1)$-dimensional subspace of $\boldsymbol{a}^\perp$. By the rotational invariance of the uniform distribution on the Grassmannian $\text{G}(n-1, N-1)$, this follows a Beta distribution:

$$\|\boldsymbol{P}_{U \cap \boldsymbol{a}^\perp}(\boldsymbol{b}_\perp)\|^2 / \|\boldsymbol{b}_\perp\|^2 \sim \text{Beta}\left(\frac{n-1}{2}, \frac{N-n}{2}\right). \tag{57}$$

According to the Theorem B.8, we have:

$$\mathbb{P}\left(\left|\|\boldsymbol{P}_{U \cap \boldsymbol{a}^\perp}(\boldsymbol{b}_\perp)\|^2 - \mathbb{E}[\|\boldsymbol{P}_{U \cap \boldsymbol{a}^\perp}(\boldsymbol{b}_\perp)\|^2]\right| \geq t\right) \leq 2\exp\left(-c(N-1)t^2\right). \tag{58}$$

Naturally,

$$\mathbb{P}\left(\left|\|\boldsymbol{P}_U(\boldsymbol{b})\|^2 - \mathbb{E}[\|\boldsymbol{P}_U(\boldsymbol{b}))\|^2]\right| \geq t\right) \leq 2\exp\left(-c(N-1)t^2\right). \tag{59}$$

The expectation $\mathbb{E}[\|\boldsymbol{P}_U(\boldsymbol{b})\|]$ is bounded using Jensen's inequality and the moments of the Beta distribution:

$$\mathbb{E}\left[\|\boldsymbol{P}_U(\boldsymbol{b})\|\right] \leq \sqrt{\mathbb{E}\left[\|\boldsymbol{P}_U(\boldsymbol{b})\|^2\right]} = \sqrt{(\boldsymbol{a}, \boldsymbol{b})^2 + \mathbb{E}\left[\|\boldsymbol{P}_{U \cap \boldsymbol{a}^\perp}(\boldsymbol{b}_\perp)\|^2\right]}. \tag{60}$$

Table 1: **Theoretical Cosine Increase Between Test Gradients and Their $n$-dimensional Krylov Subspace Projections.**

| Dataset | ViT-B/32 & ViT-B/16 | | | ViT-L/14 | | |
|---|---|---|---|---|---|---|
| | $n$ | $N$ | $\frac{\Gamma\left(\frac{n}{2}\right)\Gamma\left(\frac{N-1}{2}\right)}{\Gamma\left(\frac{n-1}{2}\right)\Gamma\left(\frac{N}{2}\right)}$ | $n$ | $N$ | $\frac{\Gamma\left(\frac{n}{2}\right)\Gamma\left(\frac{N-1}{2}\right)}{\Gamma\left(\frac{n-1}{2}\right)\Gamma\left(\frac{N}{2}\right)}$ |
| VLCS | 10 | 94725 | 9.48e-3 | 10 | 249605 | 5.84e-3 |
| PACS | 10 | 95751 | 9.43e-3 | 10 | 251143 | 5.82e-3 |
| OfficeHome | 10 | 176805 | 6.94e-3 | 10 | 372645 | 4.78e-3 |
| TerraIncognita | 10 | 97290 | 9.36e-3 | 10 | 253450 | 5.80e-3 |
| DomainNet | 10 | 269145 | 5.62e-3 | 10 | 511065 | 4.08e-3 |

The second term relates to the mean of $\text{Beta}\left(\frac{n-1}{2}, \frac{N-n}{2}\right)$:

$$\mathbb{E}\left[\|\boldsymbol{P}_{U \cap \boldsymbol{a}^\perp}(\boldsymbol{b}_\perp)\|^2\right] = \frac{n-1}{N-1}\|\boldsymbol{b}_\perp\|^2 \leq \frac{n-1}{N-1}. \tag{61}$$

However, a tighter bound arises from the expectation of the square root:

$$\mathbb{E}\left[\frac{\|\boldsymbol{P}_{U \cap \boldsymbol{a}^\perp}(\boldsymbol{b}_\perp)\|}{\|\boldsymbol{b}_\perp\|}\right] = \mathbb{E}\left[\sqrt{Y}\right], \tag{62}$$

$$Y \sim \text{Beta}\left(\frac{n-1}{2}, \frac{N-n}{2}\right). \tag{63}$$

For $Y \sim \text{Beta}(\alpha, \beta)$, $\mathbb{E}[\sqrt{Y}] = \frac{\Gamma(\alpha+\frac{1}{2})\Gamma(\alpha+\beta)}{\Gamma(\alpha)\Gamma(\alpha+\beta+\frac{1}{2})}$. Substituting $\alpha = \frac{n-1}{2}$, $\beta = \frac{N-n}{2}$:

$$\mathbb{E}\left[\|\boldsymbol{P}_{U \cap \boldsymbol{a}^\perp}(\boldsymbol{b}_\perp)\|\right] = \|\boldsymbol{b}_\perp\|\frac{\Gamma\left(\frac{n}{2}\right)\Gamma\left(\frac{N-1}{2}\right)}{\Gamma\left(\frac{n-1}{2}\right)\Gamma\left(\frac{N}{2}\right)}. \tag{64}$$

Since $\|\boldsymbol{P}_U(\boldsymbol{b})\| \leq |\langle\boldsymbol{a}, \boldsymbol{b}\rangle| + \|\boldsymbol{P}_{U \cap \boldsymbol{a}^\perp}(\boldsymbol{b}_\perp)\|$ by the triangle inequality and $\|\boldsymbol{b}_\perp\| < \|\boldsymbol{b}\| = 1$, we obtain:

$$\mathbb{E}\left[\|\boldsymbol{P}_U(\boldsymbol{b})\|\right] \leq |\langle\boldsymbol{a}, \boldsymbol{b}\rangle| + \frac{\Gamma\left(\frac{n}{2}\right)\Gamma\left(\frac{N-1}{2}\right)}{\Gamma\left(\frac{n-1}{2}\right)\Gamma\left(\frac{N}{2}\right)}. \tag{65}$$

$\square$

## B.2 KRYLOV SUBSPACE ANALYSIS

In this section, we conduct an empirical investigation into the behavior of the projection length of test gradients onto Krylov subspaces as the subspace dimension increases. This relationship is systematically examined across five diverse datasets and three distinct neural architectures, with the comprehensive experimental results visualized in Figures 1 through 5. As quantitatively summarized in Table 1, the improvement in cosine similarity remains remarkably modest ($\leq 0.01$) across all tested datasets and architectures when approximations are made via a 10-dimensional random subspace. This minimal average gain, however, contrasts sharply with the dynamic patterns revealed in the detailed subplots of Figures 1-5. For each individual experimental run and configuration, the increase in cosine similarity far surpasses the seemingly negligible average of 0.01. A particularly striking example that underscores this discrepancy is observed on the "LabelMe" domain within the VLCS dataset when utilizing the ViT-B/16 architecture, where the cosine similarity exhibits a substantial increase, exceeding 0.6. This pronounced enhancement strongly suggests that Krylov subspaces possess an inherent capability to align more closely with the true direction of test gradients, thereby offering a more effective and powerful framework for gradient approximation tasks.

## C   REPRODUCIBILITY

Our research is grounded in DomainBed [2], a comprehensive and widely-adopted benchmark suite designed for domain generalization studies, which is openly available under the permissive MIT license. DomainBed provides a standardized framework for evaluating the robustness of machine learning models across diverse data distributions, ensuring fair comparisons and reproducibility in domain shift scenarios. All experiments were conducted on a single NVIDIA Tesla V100 GPU.

## D   UTILIZATION OF LLMS

In the process of academic writing, Large Language Models (LLMs) are employed solely for language refinement purposes, including modifying grammatical errors, selecting more appropriate vocabulary, and polishing sentences to enhance clarity and flow, without being involved in any core scientific research activities such as idea generation or data analysis. This utilization leverages LLMs' strengths in text generation and syntactic understanding to improve the readability and overall quality of the manuscript while ensuring the intellectual contributions remain entirely human-driven.

## E   BROADER IMPACTS

This paper is primarily dedicated to the development of an effective domain generalization method aimed at addressing the pervasive challenge of domain shifts in machine learning systems. Given the prevalence of domain shifts and their detrimental impact on model performance with unseen data, our work focuses on improving model robustness and adaptability. By seeking models with better generalization, the proposed method reduces model bias toward spurious correlations, thereby promoting fairness and ethical alignment in automated decision-making. Consequently, we anticipate that this research will have a positive societal impact by contributing to more reliable and equitable AI systems. After careful consideration, we do not foresee any significant negative social implications arising from this work.

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

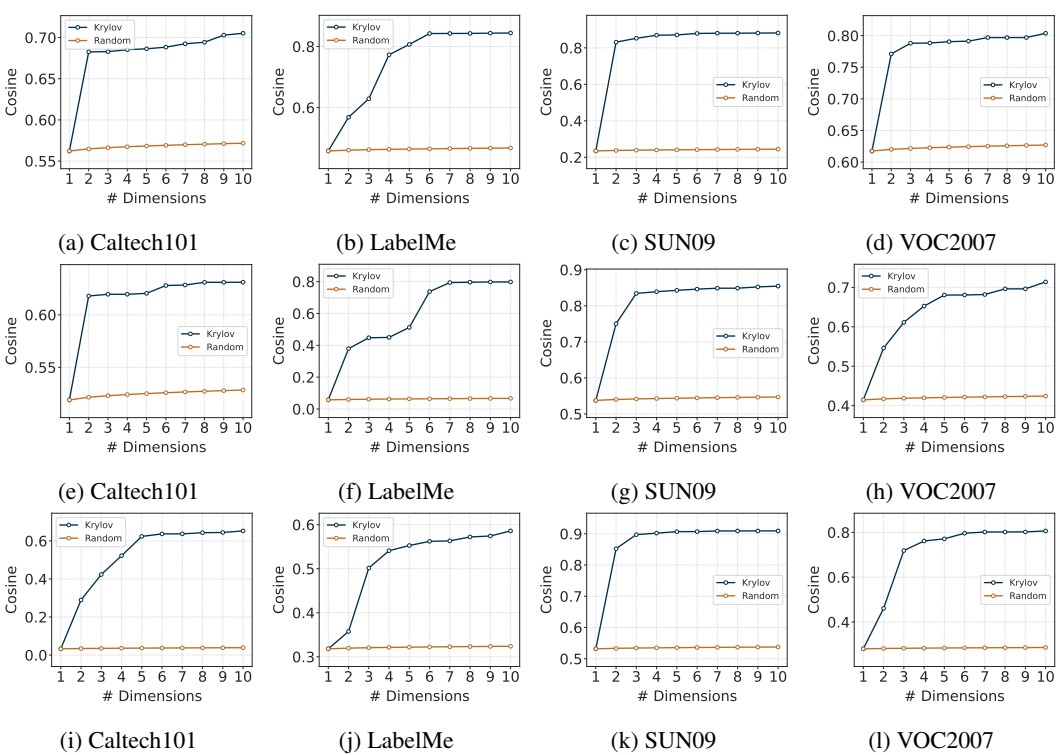

Figure 1: **Cosine Similarity Between Test Gradients and Their Krylov Subspace Projections Across Dimensions on VLCS.** The subplots in the first, second, and third rows display the performance for the ViT-B/32, ViT-B/16, and ViT-L/14 models, respectively.

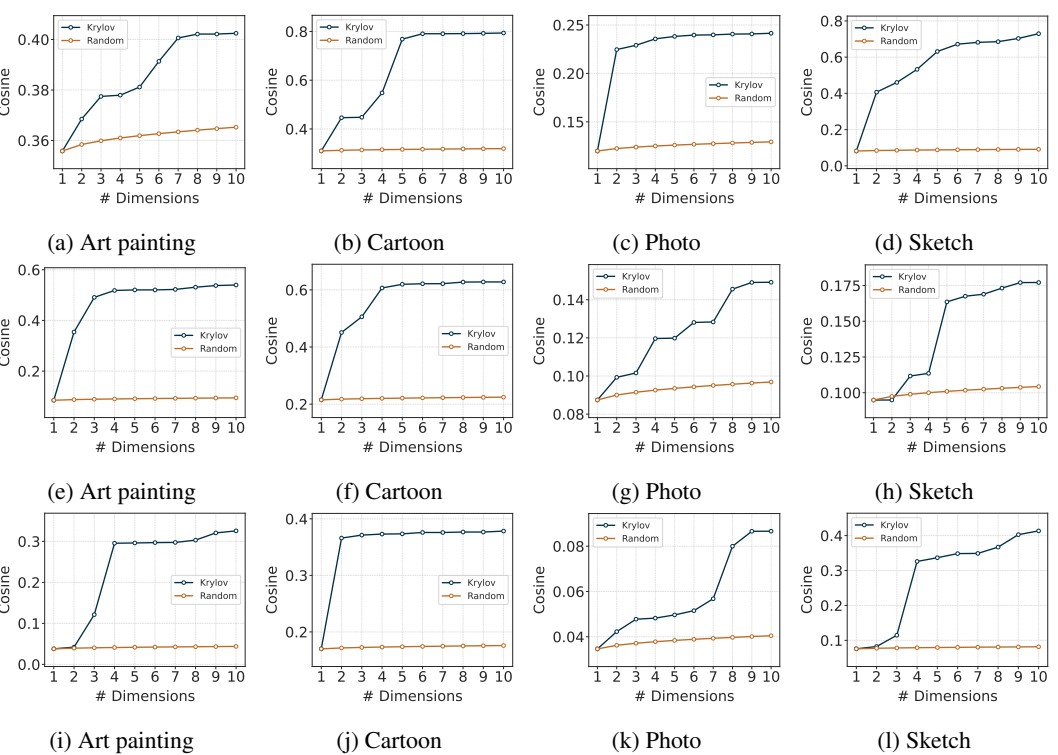

(a) Art painting     (b) Cartoon     (c) Photo     (d) Sketch

(e) Art painting     (f) Cartoon     (g) Photo     (h) Sketch

(i) Art painting     (j) Cartoon     (k) Photo     (l) Sketch

Figure 2: **Cosine Similarity Between Test Gradients and Their Krylov Subspace Projections Across Dimensions on PACS.** The subplots in the first, second, and third rows display the performance for the ViT-B/32, ViT-B/16, and ViT-L/14 models, respectively.

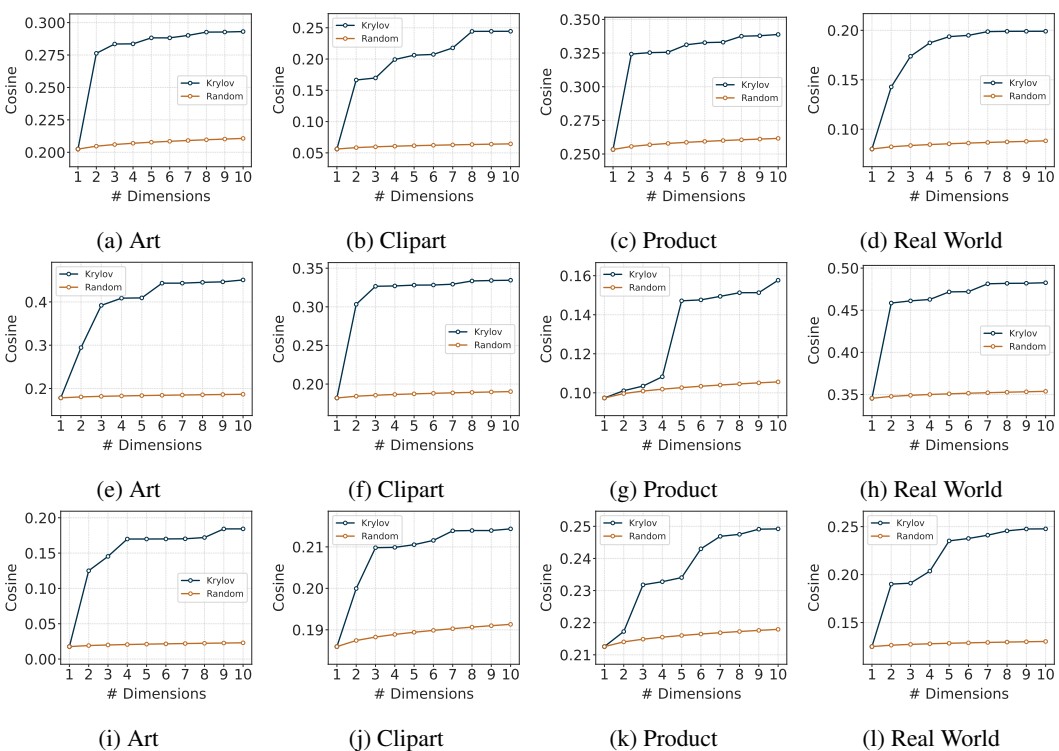

Figure 3: **Cosine Similarity Between Test Gradients and Their Krylov Subspace Projections Across Dimensions on OfficeHome.** The subplots in the first, second, and third rows display the performance for the ViT-B/32, ViT-B/16, and ViT-L/14 models, respectively.

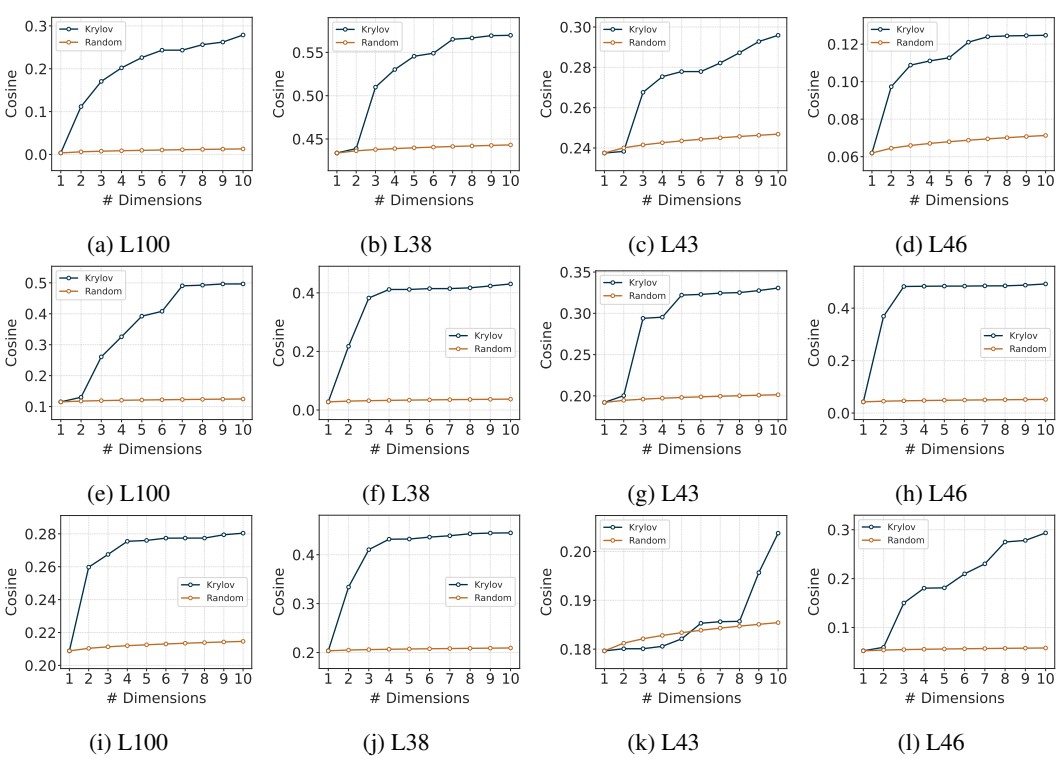

Figure 4: **Cosine Similarity Between Test Gradients and Their Krylov Subspace Projections Across Dimensions on TerraIncognita.** The subplots in the first, second, and third rows display the performance for the ViT-B/32, ViT-B/16, and ViT-L/14 models, respectively.

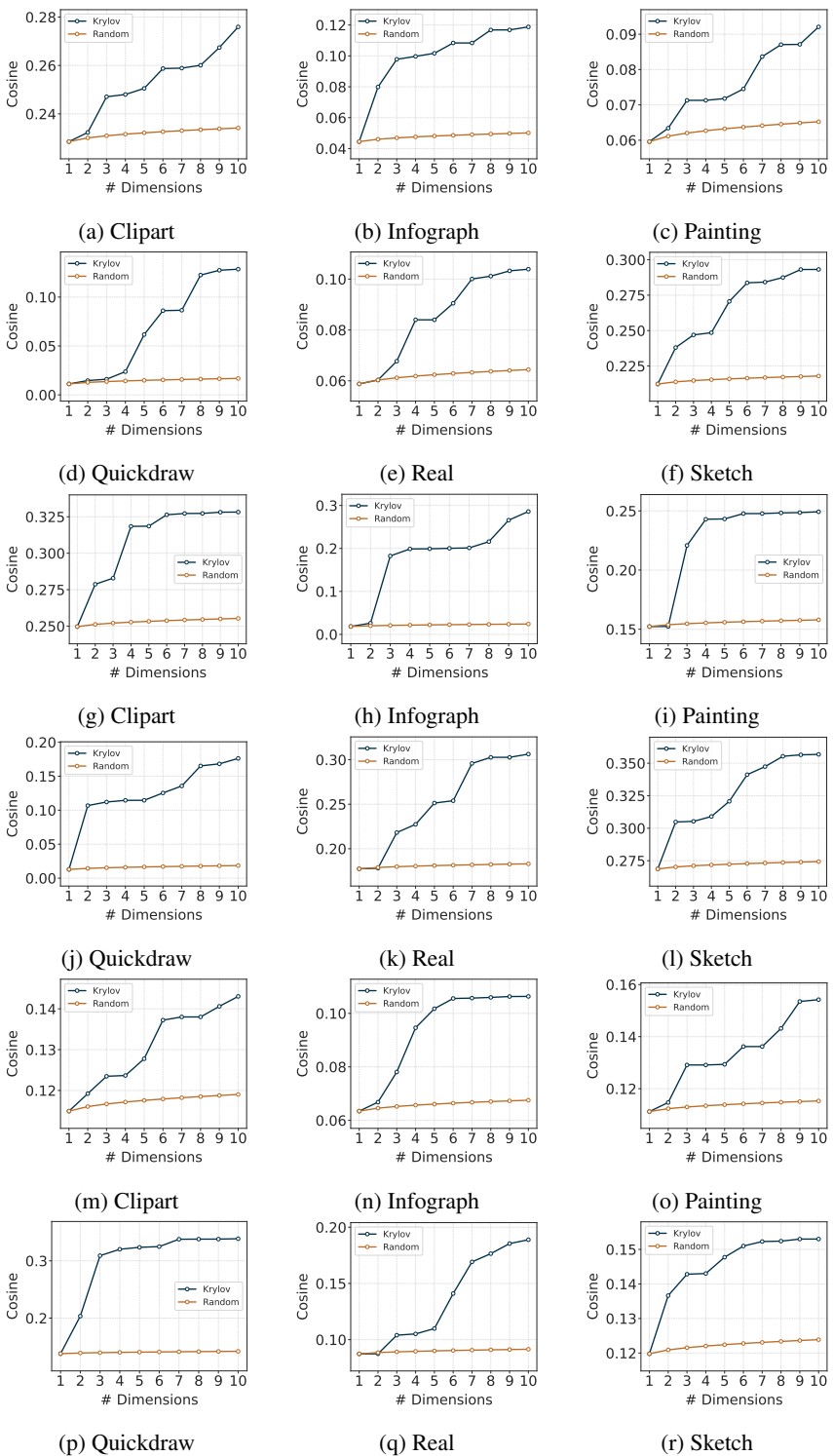

Figure 5: **Cosine Similarity Between Test Gradients and Their Krylov Subspace Projections Across Dimensions on DomainNet.** The subplots in the first two rows ((a)-(f)) correspond to the ViT-B/32 model; The subplots in the next two rows ((g)-(l)) correspond to the ViT-B/16 model; The subplots in the final two rows ((m)-(r)) correspond to the ViT-L/14 model.