# OpenReview forum: "Exploring Mode Connectivity in Krylov Subspace for Domain Generalization"
_ICLR.cc/2026/Conference — ICLR 2026 Poster_

### Official Review · Reviewer_Gedm · 2025-10-31

**Soundness:** 2
**Presentation:** 2
**Contribution:** 2
**Rating:** 2
**Confidence:** 3

**Summary:**

To find flat minima that perform better on the test data under distribution shift, the paper considers mode connectivity. It proposes a Billiard Optimization Algorithm (BOA) that traverses the flat basin of the loss landscape analogous to the reflection of a billiard on the board. To find the search direction effectively, the paper leverages the observation that test loss gradients align with the Krylov subspace. Experiments on five datasets show the potential of BOA.

**Strengths:**

1. It is meaningful to use mode connectivity to find better flat minima in domain generalization. The observation that test loss gradients align with the Krylov subspace is interesting and inspiring.

2. BOA proposes a novel strategy to search over the loss landscape.

3. Figure 6 is very helpful in understanding how BOA works.

**Weaknesses:**

1. Figure 1 suggests that minima that are equally flat on the loss landscape of the training distribution can have different sharpness on the loss landscape of test data. The observation motivates finding better minima through mode connectivity. BOA relies on a validation set of the test data to find an optimal model. However, in domain generalization, it is common to assume that test data is not accessible. My biggest concern is that under this condition, how to guarantee that the minima found via BOA are better?

2. The clarity of section 3 should be improved to aid understanding. For example, some notations in section 3 are not explained in the main paper (e.g. $\alpha, \epsilon$), and the role of $h$ and its connection with $\alpha$ are not explicitly stated.

**Questions:**

1. How is $\alpha^*$ in equation 4 determined?

2. BOA's reflection results in symmetric exploration of the loss landscape. The geometry of the loss landscape and initial search points both affect the efficiency of the symmetric search. How robust is the symmetric exploration in searching flat minima when the flat basin is not symmetric, and the initial search point lies at the center of the flat basin?

3. Does the search angle $\phi$ affect the search of optimal model? Why is it set to a fixed value instead of as a hyperparameter?

4. How is $\epsilon$ in equation 8 determined?

5. Is it a fair comparison with other DG methods, given that many of them do not utilize a validation set of test data?

---

> ### Author Response · Authors · 2025-11-22
>
> Dear Reviewer Gedm,
>
> Thank you for your thorough and insightful review. We sincerely appreciate the time and effort you dedicated to evaluating our work. Below, we provide a point-by-point response to each of the comments and concerns you raised.
>
> **Q3-1:** **How is $\alpha^\star$ in Equation 4 determined?**
>
> **A3-1:** During the line search phase, BOA identifies the loss contour boundary through line search along directional vectors, mimicking a ball approaching the boundary of a billiard table. This procedure mathematically corresponds to solving the nonlinear equation with respect to $\alpha$ (described in Equation (2)). Its approximate optimal solution $\alpha^\star$ is obtained via a golden-section search. To avoid ambiguity, we use $h$ to represent the candidate solutions/intervals of Equation (2) during the golden-section search. By the way, Algorithm 1 in Appendix A might provide additional clarity regarding BOA's implementation details.
>
> **Q3-2:** **About Symmetric Exploration.**
>
> **A3-2:** The design of BOA is inspired by the physical dynamics of a billiard system, which has been extensively analyzed in mathematical literature. Please note that billiard dynamics often exhibit ergodicity, which means this dynamic system will eventually explore nearly all of the space after many reflections. Although each individual reflection is symmetric with respect to the surface normal, it does not mean that BOA will fail to locate flat minima when the flat basin is asymmetric.  Additionally, BOA requires only that the initial search point lies within the loss contour, not necessarily at the center of a flat basin. To achieve this, a warm-start strategy, i.e., performing several SAM/ERM steps, can be employed.
>
> **Q3-3:** **About Search Angle $\phi$.**
>
> **A3-3:** We agree that the search angle $\phi$ affects the search process. Please note that $\phi$ is not set to a fixed value in our paper. Instead, it evolves dynamically according to the billiard reflection law: $\phi_{\text{incident}}=\phi_{\text{reflected}}$. At each iteration, $\phi$ is determined geometrically by the the incident angle $p_{i-1}$ and the local gradient-based normal vector. The initial direction $p_{0}$ is constructed within the Krylov subspace to approximate the oracle test gradient, and subsequent reflections adaptively update $\boldsymbol{p}_{i}$ via Equation (5). Thus, the search direction evolves throughout the optimization process. According to the ergodicity of billiard dynamics, the above evolution of $\phi$ ensures comprehensive exploration of the loss landscape.
>
> **Q3-4:** **How is $\epsilon$ in Equation 8 determined?**
>
> **A3-4:** In Equation (8), to approximate Hessian-vector products via finite differences, $\epsilon$ is typically chosen as a small value (e.g., 0.01) for effective results.
>
> **Q3-5:** **Concern about Validation Set.**
>
> **A3-5:** To ensure a fair comparison, we have already employed the same validation set in our experiments (as presented in Table 2) for BOA and other sharpness-aware methods, which are recognized as state-of-the-art DG approaches.
>
> To provide further clarity, we would like to offer the following points of clarification:
> * Model selection for DG relies on reliable out-of-domain (OOD) generalization indicators. However, neither training-domain validation set (TRVS) accuracy nor established metrics like sharpness reliably predict OOD generalization.
>   * Models along a training trajectory can achieve nearly identical TRVS accuracy while differing by more than 10% in DG performance. It suggests that a low-loss region of TRVS may contain models with widely varying generalization. Then, relying on TRVS eventually reduces to random guessing.
>
>   *  A number of studies have also revealed the limitations of existing generalization indicators (such as sharpness) in accurately characterizing generalization. In particular, a flatter minimum does not consistently lead to better DG performance.
>
> * The absence of reliable indicators for OOD generalization means we still have to use a test-domain validation set as a necessary compromise. It also suggests that model selection is still an unresolved and critical challenge in the DG community.
> * Apart from model selection, the visualization in Figure 6 demonstrates that BOA has already included models with superior generalization in its training trajectory. This phenomenon is also closely related to the concept of "grokking". In contrast, traditional DG methods often fail to reach such models within a feasible number of gradient steps. This observation further highlights the advantage of BOA and points to a promising new research direction for domain generalization.
>
> Thanks again for your detailed review and feedback. We will incorporate valuable details into the latest version later.

---

> > ### Comment · Reviewer_Gedm · 2025-11-26
> >
> > Thank you for the clarification.
> > - Based on reply A3-3, it seems that $\phi$ used in equation 3 and line 246 has a different meaning from $\phi$ used in Figure 2(c) and line 266. The notation should be updated to avoid misunderstanding.
> > - I understand that all results stated in Table 2 are selected based on train domain validation set (line 345), which means that across multiple runs, the best run is selected based on model's performance on train domain validation set. I acknowledge the limitation about model selection by train domain validation set as stated in the reply. However, my concern is that, line 271 specifically mentions that a validation set of testing is used to select an optimal model for one run. How do other methods select an optimal model for one run? Do they also use a validation set of testing?

---

> ### Author Response · Authors · 2025-11-30
>
> Thanks for your further feedback. We have carefully revised our manuscript to address your concerns. The following specific modifications have been made in response to your comments:
>
> *   **Regarding the notation:** As suggested, we have changed the symbol $\phi$ in Equation (3) to $\psi$ to prevent any notational conflict.
>
> *   **Regarding the validation set:** We sincerely appreciate your further detailed comment. To address your concern, Table 2 has been revised to ensure that all baselines are evaluated using the same test-domain validation set applied to our BOA. In fact, to ensure a fair comparison, the key baselines, ERM and sharpness-aware methods (SAM, GSAM, GAM, SAGM and DISAM), had already been evaluated using the same test-domain validation set in our original manuscript. Nonetheless, this revision creates a perfect comparison under a unified experimental setup, further strengthening the validity of our comparative analysis.
>
> Importantly, BOA maintains its performance advantage in the revised Table 2. More fundamentally, Figure 6 also provides clear evidence that this effectiveness originates from two critical phenomena: Mode Connectivity and Krylov Alignment, which were consistently observed across diverse datasets and architectures. These findings presented in this paper not only provide a novel perspective for understanding domain generalization, but also represent a paradigm shift in the principles underlying DG algorithm design.
>
> We believe these revisions and clarifications could address your concerns well. Thank you once again!

---

### Official Review · Reviewer_LAoM · 2025-10-31

**Soundness:** 4
**Presentation:** 3
**Contribution:** 3
**Rating:** 8
**Confidence:** 4

**Summary:**

This paper proposes a new optimization framework, termed "Billiard Optimization Algorithm" (BOA), to improve domain generalization using VPT with ViT backbones. This algorithm leverages the "mode connectivity" properties of the loss landscape, instead of relying only on flatness (as done by methods like SAM).

BOA consists of a line search part, where boundaries of the loss contour are reached, and a reflecion part, where the new search direction is selected using a physics-inspired rule based on the local gradients. Moreover, BOA uses the Krylov subspace to select the initial direction and constrain the search trajectory. This choice is motivated by the observation that the test gradients seem to be aligned with the Krylov subspace generated from the training gradients.

Empirical evaluation using the DomainBed benchmark with VLCS, PACS, OfficeHome, TerraIncognita and DomainNet shows that the proposed BOA method consistently outperforms the evaluated DG baselines. The paper also provides interesting theoretical analyses showing the benefits of approximating gradients using Krylov subspaces compared to random subspaces.

**Strengths:**

- This paper presents an interesting and novel optimization approach towards domain generalization introducing several new (for DG) concepts, like mode connectivity, the use of a new search approach and the use of Krylov subspaces
 - The paper also offers theoretical and empirical insights on the properties of the Krylov subspace, especially regarding its alignment with the test gradients.
 - The results presented consistently outperform the compared DomainBed baselines and sharpness-aware methods
 - Although no detailed (theoretical or empirical) analysis of the computational complexity of the proposed method is provided, the method appears to be efficient since it avoids the computation of the Hessian.

**Weaknesses:**

- The reason / underlying mechanisms behind the observed improvements remain unclear. The paper does not convincingly explain why mode connectivity or Krylov alignment lead to better cross-domain generalization.
 - The experiments are constrained to VPT with ViTs. Although this choice may indeed lead to the best results, the paper would be much more convincing if similar findings were observed e.g., for ResNet backbones and/or with full fine-tuning. Even if performance is degraded, demonstrating consistent improvements over comparable baselines would better support the stated claims. In addition, given the increased dimensionality of the parameter space, the proposed method should benefit even more compared to the baselines in this case.

**Questions:**

- How sensitive is the method to the hyperparameters (e.g., K, reflection count)?
- What is the computational cost / runtime compared to SAM or GSAM?

---

> ### Author Response · Authors · 2025-11-22
>
> Dear Reviewer LAoM,
>
> Thank you for your thorough and insightful review. We sincerely appreciate the time and effort you dedicated to evaluating our work. Below, we provide a point-by-point response to each of the comments and concerns you raised.
>
> **Q2-1:** **About the Underlying Mechanisms.**
>
> **A2-1:** This paper provides an experimental demonstration of two key phenomena: the **mode connectivity** and the **Krylov alignment**. The former suggests that it is possible to traverse the loss landscape from one initial point to discover another model with superior generalization, and the latter provides a promising direction or a constrained search space for this trajectory. Although these findings are currently empirical, they hint at profound underlying theoretical principles.
>
> * **The existence of mode connectivity likely stems from the over-parameterization of modern models.**  It can be intuitively compared to solving a system of linear equations. When the number of parameters vastly exceeds the number of constraints, the system yields a solution space (a manifold) rather than a single solution. Similarly, the low-loss region of a neural network can be viewed as the solution manifold for a highly complex, non-linear system. Well-generalizing solutions may correspond to a specific subset of this manifold. Conceptually, incorporating test-set constraints into the "system" would result in a smaller, more refined solution space, which would be a sub-region of the original manifold.
>
> * **The phenomenon of Krylov alignment may originate from a fundamental geometric similarity between the training and test data.** A necessary condition for domain generalization is that the test set shares some commonality with the training set. This implies that their respective loss landscapes might possess a similar geometric structure, such as sharing a dominant low-dimensional structure. This idea connects to Krylov subspace methods, numerical techniques designed to solve high-dimensional linear systems by projecting them onto a dominant and low-dimensional subspace that captures the most critical features. By analogy, the Krylov alignment suggests that the salient low-dimensional structure is effectively shared between domains, and thus provides a bridge for achieving better domain generalization.
>
> However, translating these intuitions into a rigorous mathematical framework presents a significant challenge, which we identify as a central focus for our future research.
>
> **Q2-2:** **Further Experiments with Other Backbones.**
>
> **A2-2:** We greatly appreciate this suggestions about further experiments with other backbones. Please refer to **A1-1 to Reviewer mWsD** for detailed discussion.
>
> **Q2-3:** **About Sensitivity to Hyperparameters.**
>
> **A2-3:** **Sensitivity to K.** Based on the experimental results below, BOA exhibits moderate sensitivity to the hyperparameter K (ranging from 5 to 20) across different domains. However, when K becomes excessively large, the performance of BOA may be adversely affected by the increasing dimension.
>
> | K     | 5   | 10  | 15  | 20 |
> |-------------|-------|-------|-------|-------|
> | Caltech101  | 98.7  | 99.4  | 99.7  | 99.5  |
> | LabelMe     | 75.2  | 74.8  | 76.1  | 73.8  |
> | SUN09       | 87.3  | 87.1  | 86.9  | 85.5  |
> | VOC2007     | 86.9  | 87.6  | 86.4  | 87.2  |
> | Avg.        | 87.0  | 87.2  | 87.3  | 86.5  |
>
> **About reflection count.** The reflection count determines whether we can search out models with superior DG accuracy. Across all datasets, we find that 20 reflection steps are sufficient for this purpose.
>
> **Q2-4:** **More Discussion about Computational Overhead.**
>
> **A2-4:** Please refer to **A1-2 to Reviewer mWsD** for a detailed discussion of computational overhead.
>
> Thanks again for your detailed review and feedback. We will incorporate valuable details into the latest version later.

---

> > ### Comment · Reviewer_LAoM · 2025-11-28
> > **Response**
> >
> > I would like to thank the authors for their detailed responses to the review comments. Based on these, I maintain my positive assessment of the paper.

---

> > > ### Author Response · Authors · 2025-11-30
> > >
> > > We are grateful for your positive assessment of our work. The key experimental results have now been integrated into the latest version of our manuscript.

---

### Official Review · Reviewer_mWsD · 2025-11-01

**Soundness:** 3
**Presentation:** 2
**Contribution:** 3
**Rating:** 8
**Confidence:** 4

**Summary:**

This paper introduces BOA (Billiard Optimization Algorithm) for domain generalization. The idea is to stay inside a low-loss region of the training loss, move to the loss boundary with a line search, then reflect and keep going. To avoid getting lost in high dimensions, the method searches only inside a Krylov subspace built from training gradients/HVPs. The authors also show that test gradients tend to align with this subspace, and that a train-computed path often also looks good on the test landscape. On DomainBed, BOA beats ERM/SAM and several DG methods; for example, on VLCS with a ViT-B/16 model, BOA improved accuracy by 3.6 percentage points compared to SAM.

**Strengths:**

- Clear formulation and intuition. BOA’s use of a training-loss sublevel set, a concrete line-search to the boundary, and a reflection update is straightforward and well motivated.

- Low-dimensional search. Constraining the trajectory to a Krylov subspace is a sensible way to capture salient curvature directions without exploring the full parameter space.

- Useful visual evidence. Overlaying the same trajectory on train and test landscapes helps illustrate why a train-computed path can still navigate good regions on test.

- Strong empirical results. Consistent improvements on DomainBed with ViT backbones (including VPT) and a notable margin over SAM on VLCS.

**Weaknesses:**

- Limited backbones: Experiments are mostly on ViT. Results on CNNs (e.g., ResNet) or other architectures would strengthen generality.

- Unclear compute cost: Please compare total elapsed (wall-clock) time, memory, and counts of line-search/HVP calls to ERM/SAM under the same settings.

- Heuristic initial direction: The current choice is simple; more analysis or alternatives would improve justification.

**Questions:**

Is it acceptable not to include CNN backbones, or can you provide BOA vs ERM/SAM results on a standard CNN such as ResNet-50 under the same budget with brief notes on hyperparameter transferability and compute cost?

---

> ### Author Response · Authors · 2025-11-22
>
> Dear Reviewer mWsD,
>
> Thank you for your thorough and insightful review. We sincerely appreciate the time and effort you dedicated to evaluating our work. Below, we provide a point-by-point response to each of the comments and concerns you raised.
>
> **Q1-1:** **Further Experiments with Other Backbones.**
>
> **A1-1:** Thank you for this valuable suggestion to evaluate our method with CNN backbones. We are pleased to report that our experimental results lead us to two main conclusions.
>
> **First, additional experiments using a ResNet50 backbone confirm that the phenomena of Mode Connectivity and Krylov Alignment still exist in this architecture.** The superiority of Krylov subspaces over random subspaces, similar to what was shown in Figure 3, remains consistent across various datasets. As demonstrated in the table below, Krylov alignment (measured by $\cos γ_K$) even exceeds 0.75 for $K=10$ on the SUN09 dataset, representing an improvement of approximately +0.5 over the $K=1$ case.
>
> | Domain | #Dim | 2 | 4 | 6 | 8 | 10 |
> | :--- | :--- | :--- | :--- | :--- | :--- | :--- |
> | Caltech101 | Random | 0.3121 | 0.3122 | 0.3124 | 0.3125 | 0.3125 |
> | **Caltech101** | **Krylov** | **0.3317** | **0.3339** | **0.3360** | **0.3381** | **0.3381** |
> | LabelMe | Random | 0.4063 | 0.4064 | 0.4065 | 0.4066 | 0.4067 |
> | **LabelMe** | **Krylov** | **0.4937** | **0.5235** | **0.5724** | **0.5825** | **0.5855** |
> | SUN09 | Random | 0.2653 | 0.2654 | 0.2655 | 0.2657 | 0.2657 |
> | **SUN09** | **Krylov** | **0.4199** | **0.7547** | **0.7603** | **0.7610** | **0.7617** |
> | VOC2007 | Random | 0.1395 | 0.1396 | 0.1397 | 0.1399 | 0.1399 |
> | **VOC2007** | **Krylov** | **0.1406** | **0.2991** | **0.3260** | **0.3414** | **0.3707** |
>
> **Second, our BOA method maintains superior performance when implemented with a ResNet50 backbone.** As shown in the performance table, BOA achieves the highest accuracy across all four domains with an average accuracy of 81.8%, outperforming both ERM (79.0%) and SAM (80.2%).
>
> | Method | Caltech101 | LabelMe | SUN09 | VOC2007 | Avg.   |
> | :----- | :--------: | :-----: | :--: | :-----: | :----: |
> | ERM    |    98.2    |  67.1   | 73.0 |  77.6   |  79.0  |
> | SAM    |    99.6    |  65.7   | 75.1 |  80.5   |  80.2  |
> | BOA    |    **99.7**    |  **69.0**   | **77.2** |  **81.1**   | **81.8** |
>
>
> In our ResNet experiments, we found that most hyperparameters could be directly transferred from the ViT-B/16 experiments. However, due to differences in loss landscape geometry, we discovered that maintaining the step size $h=10$ (as used in ViT) would require excessive line search iterations to identify the solution interval. Therefore, we adjusted the step size to $h=1$ for the ResNet experiments to ensure efficient optimization.
>
> **Q1-2:** **More Discussion on Computational Overhead.**
>
> **A1-2:** We provide a detailed analysis of the computational overhead below. All experiments were conducted on a single NVIDIA Tesla V100 GPU under consistent settings to ensure fair comparison, using the VLCS benchmark ("LabelMe" as the test domain) and a ViT-B/16 backbone.
>
> BOA mainly involves several steps that contribute to its runtime: Initially, the Hessian-vector product (HVP) is called 20 times; For each step, line-search operations are invoked 5 to 8 times. Notably, benefiting from Krylov subspace, BOA achieves high search efficiency only with a modest increase in memory usage. As presented in the table, BOA attains the best DG accuracy of 75.6% with a runtime of 1 hour and 26 minutes and a GPU memory of 7.07 GB.
>
> | Method | Runtime (hours:minutes) | Memory (GB) | DG Accuracy (%) |
> | :--- | :--- | :--- | :--- |
> | **ERM** | 1:04 | 6.78 | 68.6 |
> | **SAM** | 1:35 | 6.78 | 69.0 |
> | **BOA** | 1:26 | 7.07 | 75.6 |
>
> Importantly, traversing the loss landscape (from one initial point to discover another model with superior generalization) is also closely related to the concept of **"grokking"**. Traditional DG methods often fail to achieve this even after an impractically large number of gradient steps (e.g., $10^7$). This further highlights the significant advantage of BOA.
>
> **Q1-3:** **Concerns on Heuristic Initial Direction.**
>
> **A1-3:** Thank you for the valuable suggestion to conduct further analysis for selecting an improved initial direction. Please note that billiard dynamics often exhibit ergodicity, which means the system will eventually explore nearly all of the space after many reflections. In high-dimensional spaces, the number required can be prohibitively large. In these cases, a well-chosen initial direction is crucial for quickly finding a desirable region. However, this need for careful selection is reduced when working within a low-dimensional Krylov subspace, where the required number of reflections is acceptably small.
>
> Thanks again for your detailed review and feedback. We will incorporate valuable details into the latest version later.

---

### Meta-Review · Area_Chair_dqiX · 2025-12-23

**Summary:**

This paper studies domain generalization from a loss-landscape perspective and proposes a novel optimization framework, termed Billiard Optimization Algorithm (BOA), which exploits mode connectivity and constrains the optimization trajectory within a low-dimensional Krylov subspace. The paper further presents an empirical observation that test gradients are strongly aligned with training-derived Krylov subspaces, enabling effective search directions without access to test data. Overall, reviewers find the problem setting meaningful and the proposed perspective interesting, and agree that the empirical results on DomainBed are strong and consistent across multiple datasets and architectures. While some reviewers raise concerns regarding the lack of a fully developed theoretical explanation and the heuristic nature of certain design choices, the rebuttal and additional experiments sufficiently clarify the empirical validity and practical effectiveness of the approach. Taken together, the work presents a novel and useful optimization perspective for domain generalization and merits acceptance.

**Reviewer Concerns:**

Reviewer mWsD:
Concerns regarding backbone diversity and computational overhead were addressed through additional experiments and runtime analysis. The remaining questions mainly relate to the heuristic choice of the initial search direction, which does not appear to undermine the empirical effectiveness of the method.

Reviewer LAoM:
The rebuttal and added analyses improve clarity and empirical coverage. While a complete mechanistic explanation of why mode connectivity and Krylov alignment lead to improved domain generalization remains open, this limitation is acknowledged and is considered acceptable for an empirical contribution.

Reviewer Gedm:
The reviewer expresses skepticism about the causal relationship between traversal along flat training-loss regions and improved test-domain performance. While this theoretical concern is valid, the consistent empirical gains across multiple benchmarks and architectures suggest that the proposed approach is effective in practice, and the concern does not outweigh the overall contribution.

**Reviewer Scores:**

Reviewer mWsD: Likely no change.

Reviewer LAoM: Likely no change.

Reviewer Gedm: Likely no change or a slight increase.

---

### Decision · Program_Chairs · 2026-01-26

Accept (Poster)